# Towards Bio-Hybrid Energy Harvesting in the Real-World: Pushing the Boundaries of Technologies and Strategies Using Bio-Electrochemical and Bio-Mechanical Processes

Abanti Shama Afroz [1], Donato Romano [1,2,*], Francesco Inglese [1] and Cesare Stefanini [1,2,3]

1 The BioRobotics Institute, Scuola Superiore Sant'Anna, Viale Rinaldo Piaggio 34, 56025 Pontedera, Italy; abantishama.afroz@santannapisa.it (A.S.A.); francesco.inglese@santannapisa.it (F.I.); cesare.stefanini@santannapisa.it (C.S.)

2 Department of Excellence in Robotics & AI, Scuola Superiore Sant'Anna, Piazza Martiri della Libertà 33, 56127 Pisa, Italy

3 Healthcare Engineering Innovation Center (HEIC), Khalifa University, Abu Dhabi 127788, United Arab Emirates

* Correspondence: donato.romano@santannapisa.it

**Abstract:** Sustainable, green energy harvesting has gained a considerable amount of attention over the last few decades and within its vast field of resources, bio-energy harvesters have become promising. These bio-energy harvesters appear in a wide variety and function either by directly generating energy with mechanisms similar to living organisms or indirectly by extracting energy from living organisms. Presently this new generation of energy harvesters is fueling various low-power electronic devices while being extensively researched for large-scale applications. In this review we concentrate on recent progresses of the three promising bio-energy harvesters: microbial fuel cells, enzyme-based fuel cells and biomechanical energy harvesters. All three of these technologies are already extensively being used in small-scale applications. While microbial fuel cells hold immense potential in industrial-scale energy production, both enzyme-based fuel cells and biomechanical energy harvesters show promises of becoming independent and natural power sources for wearable and implantable devices for many living organisms including humans. Herein, we summarize the basic principles of these bio-energy harvesting technologies, outline their recent advancements and estimate the near future research trends.

**Keywords:** bio-hybrid systems; bio-energy sources; energy; bioengineering; microbial fuel cells; bionics

## 1. Introduction

Development of sustainable and zero-carbon-emission energy resources is considered one of the most demanding goals for the current world [1]. Within the vast field of green energy harvesting, an important one is bio-energy harvesting [2,3]. Although such possibilities were proposed much earlier [4], this started to gain prominence in the early 2000s [5–7]. This relatively new research field is opening up opportunities for producing energy from the wide range of living creatures, including microorganisms [6] and macro-organisms [8], as well as from bio-hybrid organisms [9]. The application fields of these bio-energy harvesters are equally elaborated, ranging from industrial-scale energy production [10] to environmental monitoring [11,12] and biomedical applications [13,14].

The first bio-energy harvesters reported in this review are known as microbial fuel cells (MFCs) which offer the possibility to become one of the next big industrial energy solutions [15,16]. MFCs utilize catabolic metabolism of different microorganisms on a wide range of organic substrates [17] and turn them into micro bio-reactors generating electrical energy [18]. Extensive works have been performed on summarizing its various aspects like development in electrode configurations [19–22], dedicated power management systems [23,24], and cell separators [25–27]. With the prospects of becoming a next-generation

large-scale power generation technology, they are already being used for powering up remote marine sensors [23], long-distance marine communications [28], simultaneous wastewater processing with auxiliary power supply [29], metal recovery processes [30] and in biosensors [31].

The next biochemical energy harvester that has gained a considerable amount of popularity are enzyme-based biofuel cells (EBFCs) where synthetically produced enzyme molecules are immobilized on the electrodes for performing glucose oxidation and in turn power generation [32]. This particular type falls under the bio-hybrid energy harvesters [33] and is considered the promising natural resource for powering up implants and prosthetic devices [34]. Research on these implantable fuel cells is still at the animal trial level [35]. Simultaneously, investigations have also been started on testing the applicability of such cells from external body-fluids [13].

The third and final bio-energy harvester discussed in this review is mechanical in nature. These biomechanical energy harvesters have a remarkable growing market for powering smart and wearable devices [36]. These harvesters are preferred for their non-invasive nature and easy-applicability in monitoring [37], diagnostic [38] and therapeutic [39] functions. They utilize mechanical energy produced from living animals and convert them into power solutions [40]. While their use for human application is eminent [41], they have also been used for powering other living bio-hybrid animals [42].

In this work, we aim to offer a concise, functional summary of these three promising bioenergy harvesters and their progress in the recent years.

## 2. Literature Search Method

A hierarchical survey was performed where "microbial fuel cell", "enzymatic fuel cell" and "biomechanical fuel cell" were utilized as primary keywords. At the secondary level keywords "architecture", "classification" and "applications" were used with all the three primary keywords. "Electrode" keyword was used for both MFCs and EBFCs. Keywords "exoelectrogens", "photo-reactors", "membranes", "waste management" were used only for MFCs. Similarly, "enzyme immobilization" was used for EBFCs while "triboelectric nano generator", "heap", "ankle", "knee", "foot" and "upper limb" were used for biomechanical cells. A pictorial description on the use of the keywords' hierarchy is given in Figure 1.

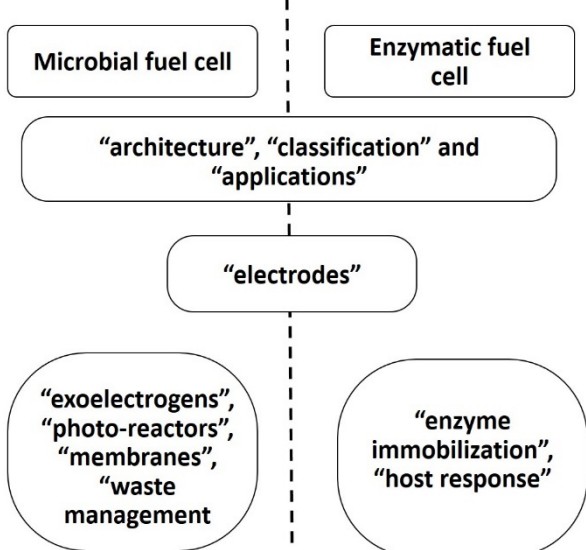

**Figure 1.** Hierarchical keywords to summarize recent progresses in bio-energy solutions.

## 3. Research Progress in Energy Harvesting from Microorganisms

MFCs extract electrical energy from different microorganisms by utilizing their catabolic metabolism over various organic components [17,18]. Bacteria are the most used microbes for this purpose, whose activity can be boosted with the aid of other microorganisms like algae [43,44].

The following paragraphs depict research progress in MFCs, their structural development, a pragmatic classification of the MFCs, adaptations performed to optimize their operations in different scenarios and their diverse conjugated applications along with electricity generation.

### 3.1. Principles of Electricity Generation with Microbial Fuel Cell (MFC)

Structurally, an MFC is composed of a cathode, an anode, the microorganism and the oxidizing substrate, which in most cases, is composed of organic matter. The structure can be of single or dual chamber type depending on the absence or presence of a separator. In a classic configuration, the microorganisms decompose the organic substrate in the anaerobic anode chamber through catabolic processes to obtain energy and generate electrons and protons/cations as a by product. The generated protons flow to the cathode chamber through the cation (permeable) exchange membranes (CEMs) and thus create a potential difference between the electrodes. The excess electrons flow from the anode to cathode via external circuit and generate a current flow [18,45,46].

A classic dual cell MFC is depicted in Figure 2. Bio-electrochemical systems started to gain attention as a possible green energy source in the early years of the 21st century [29] since microorganisms possess the flexibility to generate energy from a very wide range of biomass varieties [47]. Bacterial electron transfer mechanisms and their dependence on mediators and biofilms are three major considerations of MFC performance.

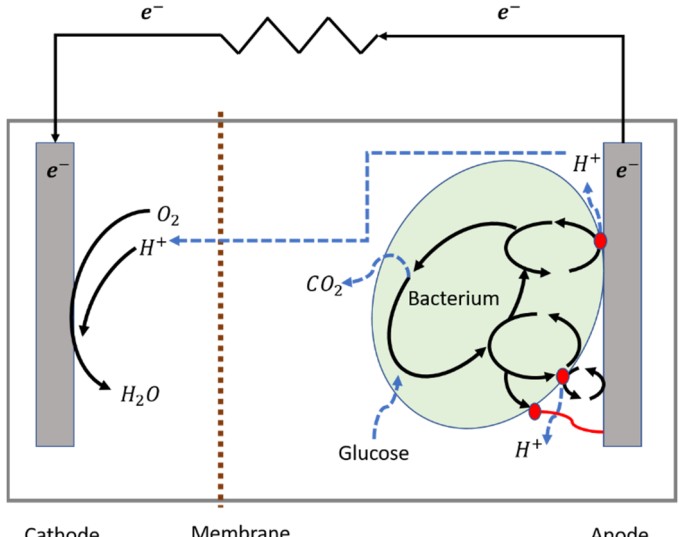

**Figure 2.** A classic dual chamber microbial fuel cell (MFC) with cathode, cation exchange membrane and anode (from left to right).

The intracellular electron transport (ET) mechanisms are of fundamental concern for MFC systems. The MFC compatible microorganisms can transfer the electrons to the anode either through direct contact or with the help of mobile electron shuttles or mediators [31,45]. These mediators are compounds that act as biocatalysts and shuttle electrons from the intracellular space to the extracellular environment within the MFC and could either be externally added or produced by the living cells [46]. The ability of microorganisms to utilize the soluble mediators as electron shuttles was highly beneficial in the early stages of MFCs. In these first-generation or mediator-dependent MFCs, presence of a suitable electron shuttle or mediator was mandatory [31]. In addition, efficient MFCs

without requiring artificial mediators also started to emerge. However, not all bacteria are suitable for generating electricity in a mediator-less configuration MFC.

Microorganisms that can effectively generate electricity in MFCs without additional mediators include a few classes of Proteobacteria in addition to some microalgae, yeast, and fungi species [48]. These microorganisms, capable of extracellular electron transfer are often referred to as exoelectrogens as well as electrochemically active bacteria, anode-respiring bacteria and electricigens [47]. They are capable of generating proteins or molecules, followed by oxidizing procedures, for transferring the electrons exogenously [48–50]. A sub group of these bacteria are nanowire generators. *Geobacter sulfurreducens* was one of the first organisms shown to produce conductive nanowires. These nanowires were latter referred to conductive pilA [48] because of their composition with the pila protein. The bacterium *Shewanella oneidensis* can also generate electrical nanowires [51,52] under special condition and as extensions of their outer-membrane, allowing the transfer of electrons to the anode without requiring soluble electron shuttles.

The presence of bacterial biofilm can be highly advantageous for MFCs because of their electroactive nature aids to generate electricity more efficiently. This biofilm is a complex, organic, polymeric matrix, produced by the bacteria themselves at any biotic or abiotic surface and these organic films can be formed by a single bacterial (pure-culture) or multiple bacterial species (mixed-culture) [48]. It has been proven that bacteria capable of forming thick anodic biofilms generate higher current densities than those who can not [53]. A pictorial summary of bacterial electron transport processes is given in Figure 3.

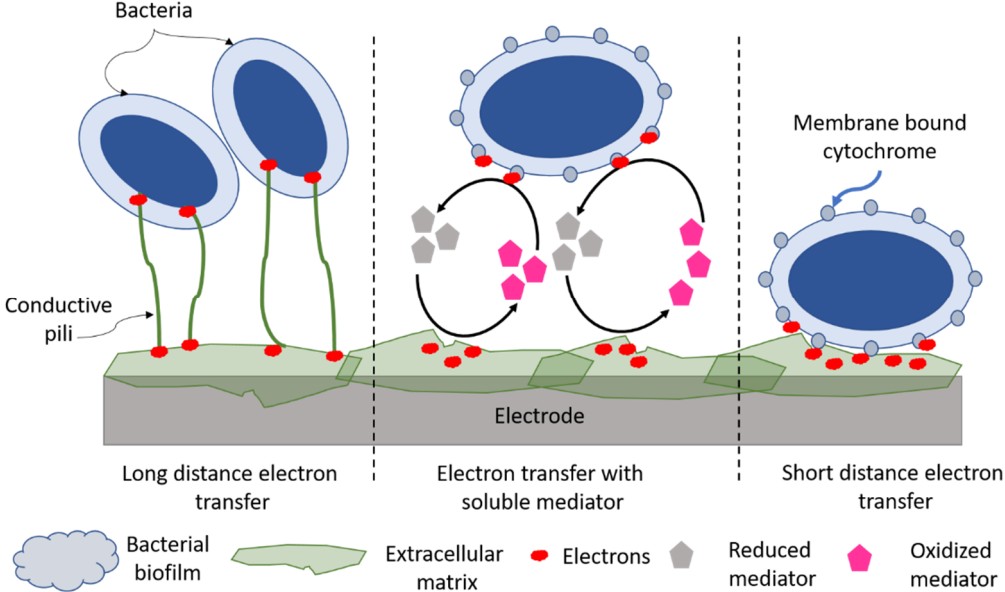

**Figure 3.** Electron transfer mechanisms, mediators and biofilm for MFC technology.

### 3.1.1. Electrodes

One of the major concerns in MFC developments has been developing efficient and economically viable electrodes [16,54]. Multiple review works have been reported with the intention of summarizing developments in MFC cathodes [19,55] as well as anodes [20,56] and in general electrode materials [21,22].

The anode accepts the electrons generated by the microbial community and hence promoting bacterial adhesion at MFC anodes is of utmost importance [57]. Additionally, ensuring an anoxic environment is required to ensure that anode is the only electron acceptor in the vicinity [58]. MFC anode materials should possess key physical features including biocompatibility [20], corrosion resistance [54], high electrical conductivity [54,59], wettability [58] and chemical strength to withstand the wastewater environment with diverse

organic and inorganic contents [56], other wastes, electrolytes and soil contaminating components.

While multiple metallic anode configurations for MFCs have been tested, stainless steel was found to be a suitable one because of its capacity to withstanding corrosion [60]. Earlier works utilized different forms of carbon anodes that provided better microbial adhesion than metal ones including carbon paper, carbon cloth, activated carbon, carbon felt, graphite felt, tungsten carbide, graphite foil and others while still not being the optimal solution because of their intrinsic hydrophobic nature [20]. Facilitating biofilm growth at the anode [48,57] has been an important consideration. It has been found that high porosity and increased surface area facilitate biofilm growth and anode surface texture also plays an important role in promoting bio-catalytic activity [61]. While biofilm growth on anode is promoted, ensuring absence of other electron acceptors except anode material itself becomes an important issue in maintaining MFCs' performance. This performance is often affected by dissolved oxygen in water that gains access to local anode spots caused by burrowing organisms [62].

Based on these research trends, we have updated the MFC anode classification based on materials by [60] into the following hierarchy as in Figure 4.

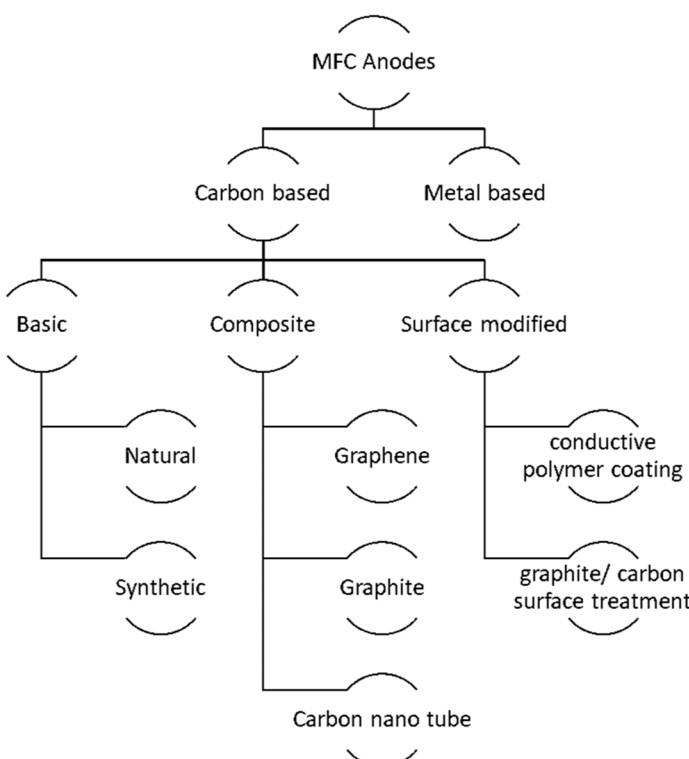

**Figure 4.** Classification of MFC anodes according to their structural configuration.

The electrons generated at the anode of an MFC cell flows through the external circuit to the cathode and completes the oxygen reduction reaction (ORR) in the presence of electron acceptors and ORR catalysts [19]. Classically, it consists of a conductive base material core and an ORR catalyst layer. Ideally, the MFC cathode should be very reactive, capable of supporting ORR catalysts as well as remaining at low cost [63], although such an optimal MFC cathode configuration has not yet been achieved. In the first generation of MFCs, expensive ORR catalysts like platinum (Pt) and copper-oxide (CuO) were widely used [61] despite their biofouling tendency and reduced capacity due to bio-poisoning caused by microorganisms [16] even in the presence of membrane [64,65]. An effective cathode configuration also requires the continuous presence of electron acceptors like oxygen in the vicinity [31] and oxygen is still the primary choice for the terminal electron

acceptor [66]. However, it has also been reported that in deep water column MFC configurations, there may appear to be an anoxic environment and in such cases, the ORR reaction is completed by other electron acceptors like nitrates, sulfates or iron oxides [67]. A classification of MFC cathode configurations is given in Figure 5.

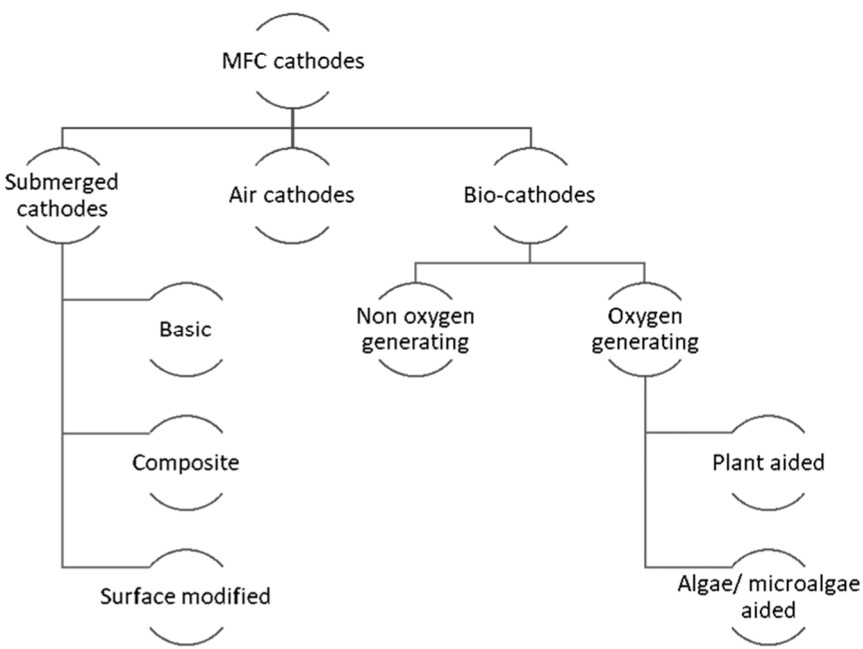

**Figure 5.** Classification of MFC cathodes according to their functionalities.

For sediment or benthic-type MFCs where submerged cathodes are used, the dissolved oxygen in water serves the purpose of electron acceptor. For such open water configurations like marine MFCs, supply of dissolved oxygen is not a big issue [6,58] and still floating marine MFCs [68] have also been implemented. For close water systems like waste-water processing plants ensuring of continuous oxygen supply becomes a common challenge [69]. Researchers have worked on providing additional air circulation at the cathodes, especially with innovative mechanical solutions. In many cases such additional units increase the production cost. While multiple examples of mechanical aeration procedures are reported, use of such units require additional cost [61,70–72]. Many works have been undertaken on improving the flexibility of such system including use of comb type [73], brush type [74] and rotating type [75] cathodes. Air cathodes [70] have emerged as a popular low-cost solution providing sustainable aeration at cathodes. In this configuration the cathode surface partially remains open to air and continuously receives oxygen supply. Janicek et al. [76] reported a generalized air cathode configuration which consists of a catalyst layer that faces the solution side of the cathode, a gas diffusion layer that faces air, and a conductive base material layer. The conductive layer also acts as a current collector as well as a mechanical support provider.

Biocathodes also became an efficient solution for continuous oxygenation requirement for MFCs. Biocathodes are defined as cathodes with attached microorganisms serving as ORR biocatalysts [77]. Biocathodes emerged as a solution where bio-fouling at cathodes was utilized as an advantage rather than a disadvantage where bacterial and micro-algae [78] were grown intentionally to aid ORR catalyst operations instead of using additional expensive catalysts. A second generation of these cathodes also include a natural oxygen generating mechanism by incorporating growth of photosynthetic bioreactors [79] on the same platform.

### 3.1.2. Membranes

In a classic dual-chamber configuration, the MFC anode and cathode are divided with a physical separator called membrane [26] where protons generated in the anode chamber travel across the membrane [80] towards the cathode chamber for the final reaction with oxygen and electrons. An optimal MFC membrane should:

1. Provide high ion, especially cation, conductivity [81];
2. Inhibit oxygen diffusion from the cathode side to the anode side for facilitating redox reaction at the cathode and maintaining anaerobic condition at the anode [82];
3. Reduce the impacts of pH slitting [83];
4. Reduce biofouling occurrence at cathode [84] and the membrane [85] by inhibiting substrate crossover [26];
5. Provide chemical stability [26].

Proton exchange membranes (PEMs) are extensively [86] used as separators in MFCS and of them Nafion is the most popular one [21,26,86]. Nonetheless, due to the excessive cost of Nafion multiple alternatives, sulfonated polymer materials [16] have been exclusively tested. Here, we propose an integrated classification of all these MFC membranes from the concepts combined form [16,27,87].

Structurally the membranes can be classified into two major groups: (i) non-porous polymer membranes and (ii) porous membranes. Non-porous membranes can be further classified into cation (CEM) and anion (AEM) exchange membranes. The famous proton exchange membranes fall under the CEM group including Nafion [88], Hyflon [89] and Ultrex [90]. These CEMs are further classified into perfluorinated membranes, hydrocarbon membranes and composite membranes. Perfluorinated membranes are of special interest as Nafion and its derivatives fall under this group. Instead of cations, AEM membranes conduct hydroxide or carbonate anions from the cathode to the anode chamber while acting as proton carriers [91]. They are preferred over Nafion and similar ones where reducing the impact of pH splitting is of importance [92–94]. Porous membranes offer higher chemical, thermal and mechanical stability and lower cost over non-porous ones [27]. They can be further classified into ceramic [95–97] and fiber types [98]. A hierarchy of MFC membranes can be seen in Figure 6.

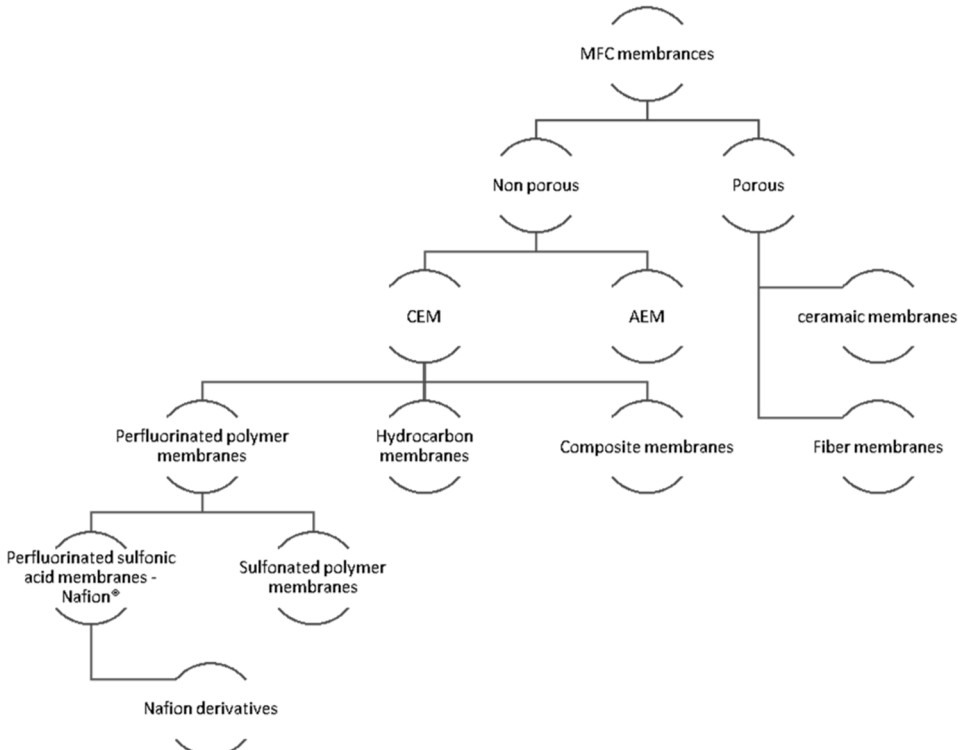

**Figure 6.** Classification of actively used MFC membranes.

### 3.2. Classification of MFCs

An MFC cell consists of an anaerobic, biotic anode component and an aerobic biotic/abiotic cathode component. Depending on the external environment, MFCs can broadly be classified into lab-scale (inside lab) and in situ (outside lab) configurations. In situ lab configurations can be further classified into aquatic and terrestrial configurations. Aquatic MFCs consist of sediment docked and non-sediment docked configurations and sediment docked configuration can be further classified into open water configurations and closed water configurations. A hierarchy of this placement dependent MFC classification can be found in Figure 7.

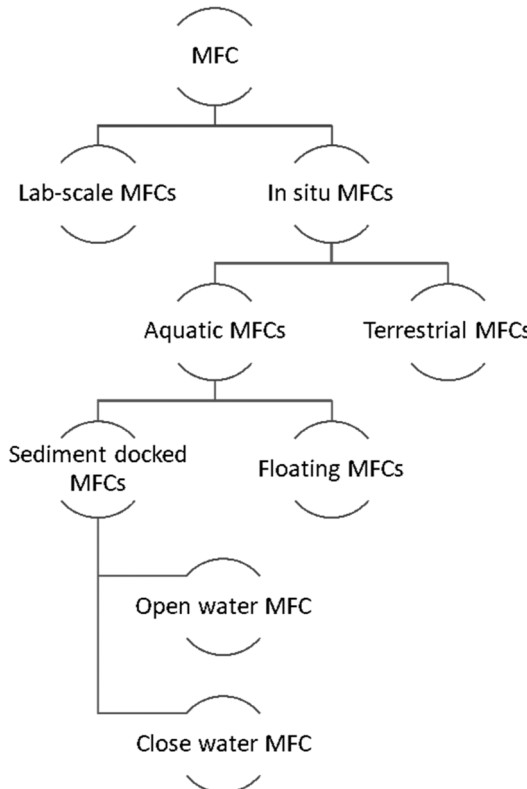

**Figure 7.** Classification of MFCs according to their placements.

### 3.2.1. Laboratory-Scale MFCs

Laboratory-scale or in-lab MFCs are smaller and are usually in the order of milliliters in volume [99–101]. These lab-scale MFCs are the most investigated ones and have been extensively utilized to identify fundamental conditions like anode-materials [102,103] and configurations [104,105], improved catalyst/cathode configurations [106,107] as well as for the study of microbial communities [101].

For scaling up operations, it is also conventional to implement a lab-scale MFC and achieve an optimal output, mimicking the desired outdoor conditions [108–110] and then transfer it to the external environment for pilot studies [99,111]. However, this methodology poses huge challenges since complex configurations for a large scale are difficult to implement [100] and increase expense. Bio-fouling and clogging become more severe in the case of in situ long-term MFC operations [99]. Thus, in multiple cases it has been found that volumetric power densities reduce with increasing MFC reactor size [112,113].

### 3.2.2. In Situ MFCs

A classic in situ MFC has a biotic anode and an abiotic cathode and is placed in an external environment outside the lab and hence requires a robust architecture. They can be further classified into aquatic and terrestrial MFCs.

### 3.2.3. Aquatic MFCs

In aquatic MFCs the anode is buried in the anaerobic soil facilitating bacterial growth and metabolism while at the aerobic cathode, either submerged in water or exposed to air, the cell redox action is completed.

Sediment docked MFCs

Benthic MFCs or sediment MFCs are possibly the most studied configuration where bacterial metabolic activity occur at anaerobic, buried anodes and therefore we refer to them in our classification as sediment-docked MFCs [114]. The resulting electrons from oxidation of organic components are received by the anode and then travel via an external circuit to an aerobic, submerged cathode [115]. The limiting factors in developing this in situ benthic MFC configuration include:

1. Low output voltage and power [116];
2. Depth dependent performance of both anode [104] and cathode [68];
3. High resistance value of mass and electron transport in sediments [117];
4. Degradation of electrode materials quality due to:
    a. electrochemical deposition,
    b. corrosion,
    c. impacts of water flow,
    d. fish grazing [118], and
    e. burrowing anodes [62].

A major section of benthic MFC based research works include development of benthic MFCs for distant marine environment [11,119,120] while others include power generation from environments like lakes [28,121], mangrove lands [122], rice paddy fields [123], aquatic ponds [124], and conjugate application of power generation and biodegradation from contaminated sediments of waste water treatments [29,125,126].

Based on the condition that the sediment for burial of the anodes could be manipulated or not, we propose to classify benthic MFCs into two more subcategories: open-water and close-water benthic MFCs.

The major feature of open-water in situ MFCs is that the anode burying underwater sediment can not be manipulated for ensuring the continuous operation of MFCs. Over the last decade this particular type of energy harvester has emerged as a viable solution for distant ocean monitoring by providing electricity to low power sensors [11] as well as communication devices with special concentration toward acoustic modems [119,120,127,128]. Additionally, they are also being used as biosensors for detecting organic carbon in sea water [129].

The selection of anode and cathode materials for marine benthic MFCs has been an important field of research so that they provide structural stability [109] as well as facilitate biofilm growth [130]. There are reports of using multiple engineered configuration of carbon materials like activated carbon fiber felt [131], granulated activated carbon [109,132], modified polyaniline-graphene nanosheets [116] and composite multi wall carbon nanotube [58,130,133] materials.

Benthic MFC electrodes can be of small [11,128] (within 4 m), medium [115,134], (between 5 to 8 m) and large (above 8 m) [127]. For the large-scale MFCs, development and deployment of large-scale graphene felt cathode/anode [108] has remained as another active research area. Different electrode configurations [132] and combinations as well as application-specific power management systems (PMSs) are also of great concern for these remote-powering MFCs. Distributed benthic MFCs are considered to be a practical solution for harsh marine environment. This is due to the fact that in case of the failure of one MFC anode/cathode, the other electrodes still remain operational and thus provide enhanced durability [62,100,105,119,132,135].

The sediment MFC is the first popular version of MFCs and in addition to open water configurations, they have also been used in closed water configurations where the

anode burial segment is accessible. In many practical configurations the anode burial sediment is further treated with additional chemicals and we define this sub-branch as close water MFCs. This category includes examples of improving power generation by the addition of silica colloid in rice paddy fields [123], cellulose in aquatic ponds [124], biomass like *Acorus calamus* leaves and wheat straw [136], iron oxide [137] in anode burial sediments as well as an experimental test of adding biochars from coconut shells into a lab-scale sediment MFC [117]. This is due to the fact that inclusion of sediment ameliorations can reduce the usual high resistance values of mass and electron transport in soil [117].

Floating MFCs

Theoretically, aquatic MFC cathodes should be closer to an open water surface for the required oxygenation and the distance between the anode and the cathode should be small enough to keep the load resistance value small. Thus, for very deep configurations, sediment MFCs whose cathodes are submerged under water might not be optimal [138] and for such cases floating cathode MFCs or in short floating MFCs emerged as a viable solution.

Floating cathodes have been used for open water MFCs. For example, a floating cathode system was developed that could be attached with a buoy and does not require fine adjustments for changing water depths [138]. A similar study [68] was performed with a floating, distributed electrode configuration that was not dependent on water column depth but was very sensitive to temperature changes. They have also been used in closed water systems like domestic wastewater treatment systems [70]. A floating biocathode system was found to be efficient for removing toxicity related to scarlet RR dye, a disperse dye extensively used for dyeing polyester fibers in textile industry [139], while generating power [140]. There are also reports of long-term monitoring of anoxic wastewater [69], biosensing of floating heavy metals [141] and harvesting energy with floating cathodes from food waste [142].

A series of floating MFC configurations were reported by Schievano et al. [143,144]. Applicability of the first configuration [143] was tested both in wastewater as well as in a natural water body. Performance of the second configuration [144] was tested for powering remote sensors and wireless data transmission. A 3D floating biocathode for overcoming the effect of oxygen depletion was tested in a lab with the sediment collected from river [145]. Massaglia et al. [146] also reported a floating MFC with carbon felt-based anodes for facilitating anodic biofilm formation in uncontrolled environments and their system found to produce electricity efficiently by using seawater both as fuel and electrolyte.

3.2.4. Terrestrial MFCs

Efforts have also been given to develop MFCs that can operate outside waterbodies and are often referred to as terrestrial MFCs [147] where the cathode was air facing, and the anode burying soil acted as both electrolyte and proton exchange membrane. Huang et al. [148] reported their work where they used an insertion-type MFC for remediation of phenol in waterlogged soil and electricity generation. Pietrelli et al. [2] reported use of a terrestrial MFC to operate a wireless sensor network for land monitoring and precision agriculture where no water–soil interface was present and the anodes were buried at 8 cm depth. A similar application was reported by Adekunle et al. [141] reported the application of a terrestrial MFC into a transportable bio-battery where they used a solid anolyte soil in addition to a small water reservoir for keeping the anodes functional. Simeon et al. [149] reported a soil-based, single chamber MFC with carbon felt electrodes where performance of the substrate (soil) was improved after urine treatment.

Another promising and emerging concept is integration of terrestrial MFCs with green infrastructure for achieving a cleaner environment in urban landscapes [150,151]. Green roofs can be of particular interest [151–153] for this purpose since in addition to electricity generation, they also provide cooling impact and thus reduce additional energy demand. An additional smart agricultural application should also be noted where

MFC integrated power generation systems are utilized for plant heath monitoring applications [154,155].

### 3.3. Impact of Other Micro and Macro Organisms in MFCs

MFCs utilize catalytic activities of microorganisms on organic substrates to generate electrical energy [18]. While use of electrogenic bacteria is the most common practice [156] for MFCs, examples of using other micro, like fungi [157], and macro organisms, like algae [72] and plants [158] are also reported.

Many works have attempted to classify microorganisms based on their use in MFCs. For example, [49] classified use of electro-active microorganisms on MFC electrodes into two major classes: bacteria, archaea, eukaryotes on anodes and bacteria, archaea on cathodes. Again, these two major classes are further divided into five classes. While Kumar et al. [48] classified microorganisms for external mediator-less configuration that included five classes of Proteobacteria, identified to date, in addition to some microalgae, yeast, and fungi. Here we have classified MFCs based on their anodic biocatalysts into the following three major classes (bacteria based, yeast-based and Archaea-based MFC) as depicted in Figure 8 and this has been abridged from the classification exoelectrogens in [49]. In the review we focused on bacteria-based MFCs and give a short overview of yeast-based MFCs.

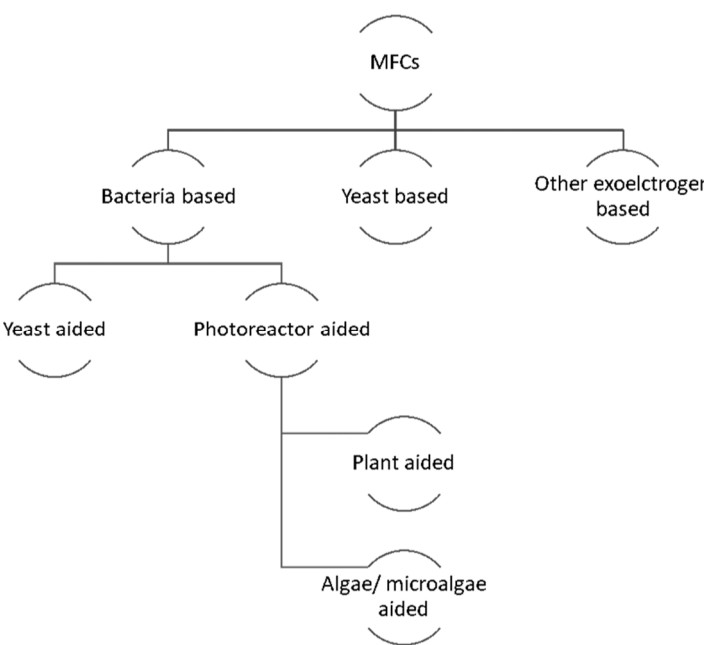

**Figure 8.** Classification of MFCs according to their catalytic activities.

Moreover, impacts of other microorganisms and the presence of other animals also impact in situ MFC performances. Erable et al. [159] hypothesized that population of electroactive bacteria attached on the marine biocathode surface had significant negative impact caused by the intervention of Protozoa, such as amoeba. Holmes et al. [160] performed multiple experiments to identify if protozoan grazing can reduce current output from sediment MFCs and identified as important the MFC current output limiting. A similar observation was also made by Al-Mamun et al. [77]. Suor et al. [161] reported their works on impact of aquatic worm predation on power generation from activated sludge with MFCs and found that it basically enhanced power output.

#### 3.3.1. Bacteria-Based MFCs

Use of bacteria for MFCS is so evident that often MFC definitions are often defined as bio-electrochemical systems which utilize bacterial metabolism to generate electrical current from a variety of organic substrates [156]. The first generation of bacteria-based

MFCs utilized artificial mediators and in the absence of suitable mediators their current production either plummeted or shut down entirely [31]. Therefore, research interest towards MFCs without artificial mediators significantly increased and still remains as an active research field often referred to as second-generation MFCs. However, only certain microorganisms are suitable for such an operation [48] and are referred to as exoelectrogens [47]. These electrochemically active bacteria are capable of extracellular electron transfer either by having conductive pili or by generating electroactive proteins or molecules that work as natural mediators [48,52] during their catabolic activity of energy extraction from biomass.

Two most studied exoelectrogen bacteria for MFCs are *Geobacter sulfurreducens* [49,162] and *Shewanella oneidensis* MR-1 [49,163,164]. *Geobacter sulfurreducens* only performs a direct electron transfer mechanism, either through the extracellular pilin can be by pili formed by PilA or appendages formed by the cytochrome OmcS [165]. The *Shewanella* species can perform both indirect electron transfer and direct electron transfer by outer membrane (OM) cytochromes mechanism within MFC. For indirect electron transfer they utilize self secreted electron mediators and direct electron transfer is performed via outer membrane cytochrome *c* and a nanowire [164]. It should be noted that nanowires in *Shewanella* are extensions of the membrane and, therefore, the direct electron transfer is also performed by OM cytochromes.

### 3.3.2. Yeast-Aided MFCs

Fungi are considered the second most commonly used microorganism for MFCs. Raghavulu et al. [157] reported his work where *Saccharomyces cerevisiae* was tested as MFC biocatalyst and their performance in the MFC setup was tested without any mediator and in three different pH conditions. The test results indicated that *Saccharomyces cerevisiae* can be used as an effective anodic biocatalyst for MFC setups. In a similar work [166] a cubic MFC reactor was fabricated with *Saccharomyces cerevisiae* PTCC 5269 as active biocatalyst with neutral red and potassium permanganate as mediators in anode and cathode compartments. Babanova et al. [46] concentrated their focus on the *Candida melibiosica* 2491 yeast strain and found that it could be used as anodic biocatalyst even without any artificial mediators. Rahimnejad et al. [167] reported that *Saccharomyces cerevisiae* can perform as an effective biocatalyst of MFC in the presence of thionine. Hubenova et al. [168] provided a review on extracellular based bio-matrices where it summarizes the top-notch yeast-based biofuel cell research and developments.

For more details on current state-of-the-art MFC systems that utilize fungi for biocatalysts please refer to the works of Sekrecka-Belniak and Toczyłowska-Mamińska [169]. Morant et al. [170] reported their primary experimental evidence that fungi isolated from the Caatinga region in Brazil can be used as efficient biocatalysts for MFCs. For their study they tested three Brazilian filamentous fungi *Rhizopus* sp. (SIS-31), *Aspergillus* sp. (SIS-18) and *Penicillium* sp. (SIS-21) and proved all three of them were compatible for MFC air-cathode configuration. Kaneshiro et al. [171] reported examining MFC power-generation performance for seven different kinds of yeast. Their performances were evaluated with a milliliter scale, dual-chamber MFC cell configuration where carbon fiber bundles were used as electrodes. They utilized a combination of glucose and xylose for substrate, which was extracted from wood biomass. They found *Kluyveromyces marxianus* to be a good candidate for biocatalytic activity and this finding can be particularly advantageous for power extraction from woody biomass. Mardiana et al. [172] reported use of yeast-based MFC system with mediator methylene blue and electron acceptor $K_3Fe(CN)_6$. Impact of two inorganic mediators, methylene blue and methyl red were tested on a fungi-MFC and was reported by Christwardana et al. [103]. The authors found that yeast *Saccharomyces cerevisiae* performs better with methylene blue. Pontié et al. [173] reported a fungus *Scedosporium dehoogii* in a MFC that can decompose acetaminophen, a widely used component in pharmaceutical industries whose degradation is quite difficult. Along with decomposing acetaminophen, the fungal MFC also provided a stable power output of 50 mW/m$^2$.

In addition to only bacteria and only fungi-based configurations, it has been proven that catalytic activities of bacteria in MFC can be boosted with addition of fungi. For example, Dios et al. [174] reported developing a combined fungus–bacterium MFC. The authors grew fungus *Trametes versicolor* with *Shewanella oneidensis* in such a way that the bacterium could use the fungus network for efficient electron transport towards the anode. Work has also been undertaken on investigating the impacts of mutual interaction of bacteria and yeasts in MFCs and the results seem very promising. For example, Islam et al. [175] found that electron shuttle-generating bacteria *Klebsiella pneumonia* boosted the performance of yeast *Lipomyces starkeyi* based MFCs.

### 3.3.3. Photo-Reactor Aided MFCs

Biocathodes for MFCs emerged as an efficient solution to the limiting factor that cathodes require presence of expensive ORR catalysts and continuous supply electron acceptors [69]. Kaku et al. [176] proposed one of the first photo-reactor coupled MFCs for paddy fields where the air cathodes got abundant oxygen supply from the green rice plants and the system works as an ecological solar cell. In line with the development of plant based MFCs, emerged algae [71,72] and microalgae [177,178] supported cathodes that provide an oxygen supply as well as acting as ORR catalysts.

There are reports of air biocathodes [179] and even floating marine biocathodes [68,159]. Chronological improvement of plant based MFCs paved the way for a newer sub-group called photo-reactor coupled MFCs. Photo-reactor coupled MFCs are very promising since they focus on cohabitation of multiple organisms, especially phototrophic ones, for efficient bio-energy production. Such photosynthetic organisms act as in situ oxygenators and facilitate cathodic reactions [180]. In addition to oxygenation, such microorganisms can also act as a regular source of substrate [79]. Rashid et al. [181] showed that algae biomass, when used with activated sludge, can be used as an effective MFC substrate component. In line with this, Campo et al. [72] reported an algae-based MFC-biocathode configuration where cathode aeration was achieved by the oxygen generated from the photosynthetic activity of the cathode itself. Luimstra et al. [182] reported a similar work on a cost-effective, MFC configuration where photosynthetic activity of cyanobacteria was utilized to improve cathode oxygenation. A similar work was reported by Gouveia et al. [178] where a Photosynthetic Alga MFC was developed for bio-electricity generation using *Chlorella vulgaris* in the cathode.

Zhang et al. [183] reported their study on bio-cathodes with aerobic microorganisms as catalysts. El Mekawy et al. [180] summarized current photo-reactor coupled MFC configurations and limitations in their scale-up operations. Lee et al. [78] wrote a mini-review on the recent studies on microalgae processes, MFC process and their coupled systems. Kabutey et al. [184] concentrated on summarizing wide applications of photo-reactor coupled MFCs. These classes of MFC can be further classified into plant and algae/microalgae based MFCs.

Rosenbaum et al. [185] first summarized the use of plant-based biocathodes to be included with MFCs. These plant-based MFCs generate bio-electricity using both living plants and bacteria [184] and offer immense possibilities for powering wetlands on a large scale [186]. The reason is that wetland plants possess the unique ability to release oxygen as well as organic matter into the rhizosphere. This principle was successfully utilized to implement a biocathode, buried in the rice rhizosphere that successfully acquired electrons from root excreted oxygen [187]. Similar plant-based MFCs have also been reported by [184,188,189]. This method is extensively used in combined constructed wetland-MFC systems [137,186,190,191]. They are also found to be more efficient in energy generation than photovoltaic cells and are also capable of removing various pollutants as well as biosensing [184].

In addition to the finding that phototrophic organisms grown on cathodes can provide additional oxygen supply [180], the finding that bacteria can also be used as biocatalysts to accept electrons from the cathode electrode emphasized the development of

bio-cathodes [192]. The closed loop of bio-mass with oxygen utilization and generation with microalgae have been intensively studied [78]. Works of Mohan et al. [193] and Gajda et al. [71] also prove this observation. Microalgae-based MFCs are capable of removing CO2 from air and nitrogen and phosphorus contaminants from waters [78,180]. Logroño et al. [194] utilized this principle for simultaneous biodegradation of textile dye waste and generation of bioelectricity and obtained promising results. Wu et al. [195] reported an MFC with a tubular photo-bioreactor configuration where *Chlorella vulgaris* was added to the cathode chamber to ensure continuous oxygen. Juang et al. [196] compared electricity generation dependency of different light power for algae-MFCs. Lobato el al. [44] tested the self-activation performance of an algae-MFC and achieved full independence.

Baicha et al. [197] reviewed the applications of microalgae based MFCs for bioenergy production. According to Saba et al. [198], a bacteria-algal MFC combination provides advantages in terms of power generation, wastewater treatment, biomass cultivation production, carbon dioxide assimilation, and oxygen production over regular MFC configuration.

### 3.4. Applications of MFCs

In the following sub-section we summarize the wide application field of MFCs. A distribution of MFC applications can be seen in Figure 9.

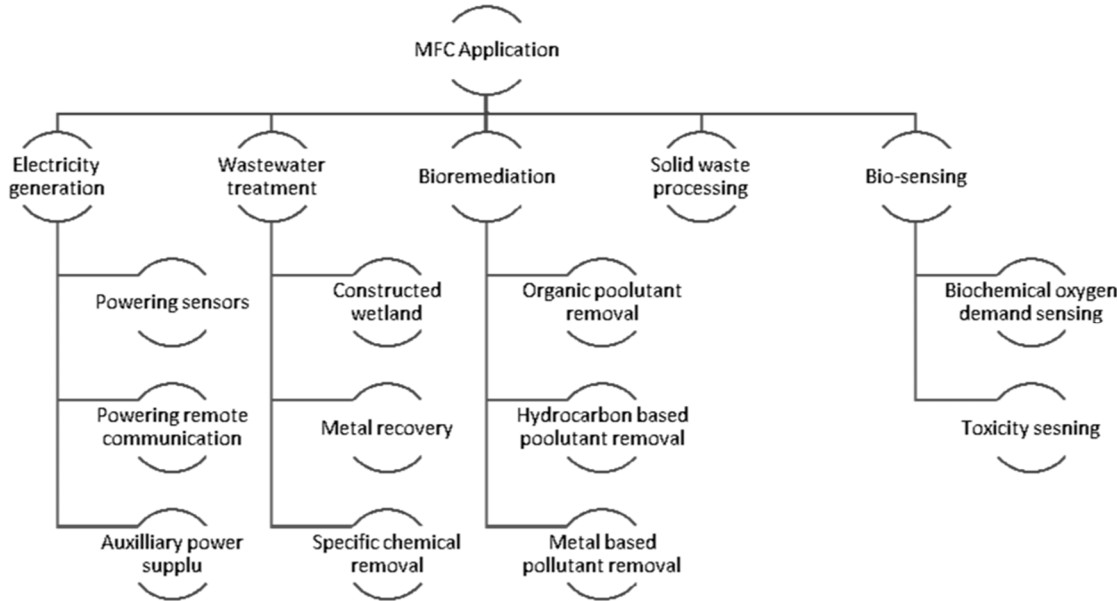

**Figure 9.** A hierarchical presentation of MFC applications.

### 3.4.1. Electricity Generation

Lab-scale MFCs, which are in the order of milliliters in volume [99] generate electricity in the range of mW/m$^2$ power density [11,102,136] and around mA/m$^2$ in current density [199]. Up until now, most of the dedicated PMSs have been developed for marine remote sensors with low power requirements [200].

MFCs are widely used for powering marine sensors which mostly measure temperature [11,23,28], levels of the phreatic aquifers [154] or pH level [201]. These applications require low power and still providing power output of this order with MFC systems is challenging. This is related to the fact that voltage reversal occur while stacking MFC cells and every MFC-based system require customized power management units. Liu et al. [202] reported cascading 18 MFCs into an array into a new type of distributed benthic MFC with dedicated PMS for powering up 2 sub-sea temperature sensors. A similar task for powering up 2 temperature sensors was performed by Khaled et al. [203] where they used a power management unit (PMU) with a fly-back converter to ensure continuous power

supply. Chailloux et al. [204] reported application of an economical off the shelf power management unit to generate micro-watt range power in both continuous and burst mode.

The popular choices for MFC PMSs are capacitors [205], charge pumps [23], boost [206] and flyback converters [207,208]. When using capacitors, there are two modes of operation: intermittent energy harvesting (IEH) [23] where energy is first stored and then released in bursts and continuous energy harvesting (CEH) mode [205] where a continuous output power is maintained at the load side. IEH mode has been proven to be more suitable for MFC-based power generation systems [112]. The most simple form of an MFC PMS is an IEH mode capacitor circuit that charges one or more capacitors until enough energy is accumulated for powering up the load. While its operation is simple and straightforward, the output voltage is limited by the open circuit potential of the capacitor used [200].

Structurally, charge pumps are slightly more complex than capacitors and yet found at low cost and are widely used in MFC PMS [23], which is basically an inductor-less direct current/direct current (DC/DC) converter with capacitors. The S-882Z series charge pump from Seiko Instruments are very popular for MFC applications [200]. A DC/DC boost converter is a customized electric circuit that converts direct current power from one voltage level to a higher level [200]. An example of popular ones for MFCs include L6920DB from STMicroelectronics [200] and LTC3108 from Texas Instruments [206,209]. Flyback converters [208] are more complex in the sense they use a coupled inductor instead. Theoretically their voltage gain can be infinite but are only applicable for below 100 W power options [210].

A recent addition to MFC PMS development is application of maximum power point tracking technology [211]. Many other components including voltage balancing circuits [122], super-capacitors [11,28,212] and semiconductor devices [109] have also been used.

For more information regarding a summarized version of recent progresses on MFC based energy harvesting systems and their dedicated PMSs, readers are referred to the reviews by Wang et al. [200]. For the recent progress on dedicated power management racking systems, please refer to the works of Abavisani et al. [211].

Many efforts have been made to customize MFC-provided electricity for remote communication purposes. In one work by Zhang et al. [28] the authors powered up a wireless temperature sensor for consistent data transmission to a PC from the sediment of Lake Michigan. Another wireless-sensor network powering by sediment-MFC was reported by Thomas et al. [209] where the implemented system provided a stable performance in sending high rate a signal from the source. Tommasi et al. [213] reported powering both a piezo-resistive pressure sensor as well as an ultra-wide-band transmitter with a 0.34 L MFC energy supply system. Schrader et al. [120] reported their works on use of marine benthic MFC that powers up sensors and acoustic modems for data transmission over a range of 0.5 km.

Even though an MFC is a promising green technology, its low voltage and power output is considered as its biggest limitation. Sediment MFCs are criticized for their rather impractical architecture for generating a reasonable amount of required energy [214].

### 3.4.2. Wastewater Treatment

To date, conjugated power generation and waste-removal has remained as one of the most popular applications of MFCs. One motivation behind this solution is that the combined system produces less sludge than conventional methods of waste treatment while generating additional power [29] that can either partially or completely meet the power requirements of the waste processing mechanism.

Jia et al. [125] reported their work on power generation with single chamber MFC with brush anodes and carbon cloth-based cathodes while processing food waste and found that a different order of organic waste loading affects the MFC performance. Tee et al. [126] reported an MFC-adsorption hybrid system with air-cathode and with granulated activated carbon based anodes. For a more detailed review of MFC application on wastewater pro-

cessing, please see the review study by Gude [215] and for the impact of wastewater substrate composition, please see the works of Pandey et al. [216]. To go into more detail of how to integrate wastewater processing with MFC technologies, please see the works of He et al. [55]. Do et al. [29] classified conventional wastewater treatment-MFC power-generation systems into 5 major groups including sediment MFCs, constructed wetland MFCs, membrane bioreactor MFCs (MBR-MFCs), desalination MFCs (DS-MFCs) and others. The authors also performed comparison between the classes in terms of their substrate, power density, and chemical oxygen demand rate.

Logroño et al. [217] reported their work on simultaneous electricity production real dye textile wastewater processing. Further work on textile waste water processing with different MFC configurations has also been reported. For example, decolorization of azo dye with biocathode configuration [218–220]. Use of algae in MFC systems has been another promising trend, whether being used as biocathode material [217] or as substrate [10].

Constructed wetland is a man-made wetland used for organic degradation for the wide range of agricultural to industrial wastewater [221]. Various efforts can be found on integrating the organic process of constructed wetland with MFCs. Oon et al. [221] built an up-flow constructed wetland where anaerobic and aerobic regions were naturally developed in the lower and upper bed and the system obtained a 100% chemical oxygen demand removal efficiency. Yadav et al. [222] reported the use of another vertical flow constructed wetland system to remove different dye from synthetic wastewater and generate electricity as well. The configuration achieved the maximum value of 93.15% dye removal efficiency following 96 h of treatment from the wastewater with 500 mgl$^{-1}$ initial dye concentration. Villaseñor et al. [223] tested the applicability of horizontal subsurface flow constructed wetland for performing simultaneous organic waste processing and power generation. This study offered two major observations: the photosynthetic activity of the macrophytes, *Phragmites australis* was dependent affected on the light/darkness changes, and this caused voltage fluctuations and affected stable performance of the system. Liu et al. [224] demonstrated that use of *Ipomoea aquatica* plant in their constructed wetland MFC system provided a higher power density and nitrogen removal in comparison to their contemporary systems. The authors also worked on optimizing the vertical constructed wetland with 3 different electrode materials (stainless steel mesh, carbon cloth, granular activated carbon) and found both stainless steel mesh and granular activated carbon's suitability for such configurations. Fang et al. [219] reported another successful combination of *Ipomoea aquatica* plantation constructed wetland-MFC system for azo dye decolorization. The planted system achieved the decolorization rate of maximum 91.24% with a voltage output of about 610 mV. Additionally, the system promoted growth of *Geobacter sulfurreducens* and *β-Proteobacteria* while inhibited Archaea growth in anode. Srivastava et al. [225] found from their study that a coupled constructed wetland-*β* MFC system performs better in removing organic substances than normal constructed wetland systems. For a more detailed review of coupled constructed wetland and MFC system, please see the review by Doherty et al. [190,226]. In [190], the authors particularly stressed the importance of maintaining anaerobic anode and oxygenated cathode configuration separated. Corbella and Puigagut [226] indicated that constructed wetland systems naturally offer aerobic conditions in the upper layers and anaerobic in the deeper ones and results in a favorable environment for MFC power generation system implementation. Xu et al. [191] identified high internal resistance as one of the limiting factors of such coupled systems. The researchers tested a new strategy, called capacitator engaged duty cycling, with an open air bio-cathode constructed wetland MFC system and obtained 19.81% more electric charge than the conventional continuous loading system.

MFC integrated wastewater treatment processes have been used to remove sulfur [227], sulfide [106,199,228,229], nitrogen [230], chromium [10] and salt components [172]. While metal contaminated waste possess great health and environmental risks, it also provides possibilities of precious metal recovery [231]. Combination of MFCs and microbial electrosynthesis have emerged as a method of choice for such metal recovery systems [30].

Fradler et al. [232] reported that heavy metal ions (for example $Zn^{2+}$, $Ni^{2+}$, $Cr^{2+}$, $V^{5+}$ or $Co^{2+}$) are often present in industrial wastewater and can be extracted using similar combined configurations. Li e al. [233] reported a self-sustained combination of MFC and microbial electrosynthesis capable of extracting three heavy metals including chromium (Cr), lead (Pb) and nickel (Ni).

### 3.4.3. Bioremediation

An additional application of MFC technology has been bioremediation and this is also one of the most investigated applications of sediment MFCs [234]. In this process microorganisms are utilized to treat polluted sites to break down environmental pollutants, to regain their original condition [235]. This has long remained as an alternative natural process of waste removal from land [236]. For more details on recent developments on bioremediation of sediments, please refer to the works of [67]. There have been particular examples of this in removing in removing organic [237] hydrocarbon [238] and metal [239] based pollutants. MFCs have been reported to recover Ag(I), Au(III), Co(II), Cd(II), Cr(VI), Cu(II), Hg(II), Pb(II), Se(IV), V(V), U(VI), and Zn(II) [30]. Yun-Hai et al. [240] reported their study on silver recovery from silver alkaline wastewater and simultaneous electricity production in a dual chamber bio-electrochemical cell.

In their review, Dominguez-Benetton et al. [241] summarized the latest mechanisms of metal recovery using MFCs. In another review, Wang et al. [231] classified mechanisms of metal recovery using MFCs in 4 different categories: direct metal recovery with abiotic cathodes; metal recovery with externally powered abiotic cathodes, metal conversion with bio-cathodes and metal conversion with externally powered bio-cathodes supplemented by external power sources.

### 3.4.4. Solid Waste Processing

Mohan et al. [242] reported a solid phase MFC system, developed to evaluate the potential of bioelectricity production by fermentation of food waste that gave promising results. The configuration utilized open air cathode sediment MFC configuration with graphite electrodes. They identified distance between the electrodes and PEM had a significant influence on power output, and the amendment of sodium carbonate improved system's power buffering capacity. Lee et al. [243] reported another MFC-based system for handling solid wastes as a feedstock. The authors evaluated the system with two configurations: (1) a single chamber, combined membrane-electrode configuration; and (2) a dual chamber, proton-membrane-less configuration with brush-type anode and double air cathode. They have used cow manure for feedstock and the second configuration provided better results with higher power output. Wang et al. [244] also reported a solid state MFC for processing cow manure with a single-chamber, air-cathode MFC configuration. The authors reported that a moisture content higher than 80% was suitable for current generation. Moreover, an addition of small amount of platinum catalyst improved the power density by 10-fold and output voltage by twice as much. Damiano et al. [245] reported their study of feasibility analysis of two MFC-based electricity generation configurations that could simultaneously treat municipal solid waste landfill leachate. They identified that for such cases, smaller configurations perform better in power generation. Pendyala et al. [246] tested the practicality of using solid municipality waste as substrate for an MFC-based system where they categorized the organic waste components in 3 main groups including food waste, paper–cardboard waste and garden waste and concluded that the organic fraction of municipal solid waste is a promising feedstock for MFC-based waste processing. Detailed data analysis from their observations indicate that the microbial composition of the anodic biofilm became a function of the feed composition. Moreover, they also found that regulating the protein content and removing furfurals and phenolic compounds from feedstock could increase the percentage of chemical oxygen demand removal rate.

### 3.4.5. Biosensing

A biosensor can be defined as a system that utilizes specific biochemical reactions to identify the presence, absence or concentration change of chemical components by electrical, thermal or optical signals [31]. The MFC current output is directly related to the metabolic activity of the electroactive microbial community in the anode zone [247], thus making the system readily usable for multiple biosensing applications. Multiple review works also targeted summarizing biosensor applications of MFCs [31,192,247–251].

Two most common applications of this category are biochemical oxygen demand (BOD) sensing and various toxicity sensing [31,248,249]. Jiang et al. [252–254] reported a copper, i.e., Cu (II) toxicity sensor, a formaldehyde detection sensor and an overall water-quality monitoring system including organic matter, heavy metal (Cu2+), and acidic toxicity detection. Quek et al. [129] reported an intuitive MFC-based biosensor configuration for detection of biofouling occurring at reverse osmosis membranes of marine desalination systems. Zhou et al. [255] reported an MFC-based carbon monoxide detecting biosensor depending on the hypothesis that carbon monoxide inhibits bacterial activity in the anode would also decrease electricity.

### 4. Research Progress in Energy Harvesting from Enzyme-Based Biofuel Cells (EBFCs)

EBFCs have emerged as the most practical form of chemical energy harvesting technology [33,256] for living organisms in implantable form and are also referred to as glucose biofuel cells. These EBFCs are the second largest group within the bioenergy harvesters. Similar to MFCs, they also depend on the electro-catalytic activity of enzymes for energy generation [4]. However, by contrast with MFCs, these cells are not intended for industrial energy production and their focus is on powering up various micro-scale [7], biomedical devices [34] that are used for clinical purposes.

### 4.1. Principles of Electricity Generation from EBFCs

EBFCs are bio-electrochemical cells that can extract energy from glucose and alcohol based organic substances [257]. In a classic EBFC configuration, glucose oxidase or glucose dehydrogenase (GDH) are immobilized at the bio-anode for glucose oxidation while oxygen is reduced at the biocathode using immobilized laccase or bilirubin oxidase and generating power [33].

Although their principle of operation is similar to MFCs, both anode and cathode of such bio-fuel cells are prepared by embedding pure enzymes [257]. Thus, at a lower level, EBFC electrodes are inherently different from MFC electrodes since they require the presence of specific enzymes without the presence of the microorganisms that generate them. However, from a top view, EBFCs are equivalent to a classic two-electrode MFC configuration since they perform oxidation of glucose at the anode and oxygen reduction at the cathode to generate electrical power [32] and remain connected via an external load resistance. A conceptual implementation of an EBFC is shown in Figure 10.

These enzyme-immobilized electrodes are an indispensable part of EBFCs where synthetically produced enzyme catalysts are assembled on electrode surfaces [259]. The development of enzyme immobilization techniques has remained as an active research area for the last 20 years [25]. Common enzyme immobilization techniques include enzyme covalent binding or "wiring", sandwiching between a protective polymer layer and the electrode surface and entrapment inside the polymeric membrane coating of the electrode [260]. This technology offers an opportunity to miniaturize the cell structure since the configuration does not require separation of fuel and oxidant [259]. A pictorial summary of various immobilization techniques has been given in Figure 11.

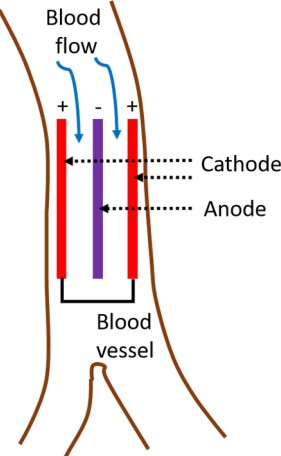

**Figure 10.** A futuristic concept of a biofuel cell for implementation within a blood vessel with enzyme immobilized electrodes, redrawn from the schematic shown in [258].

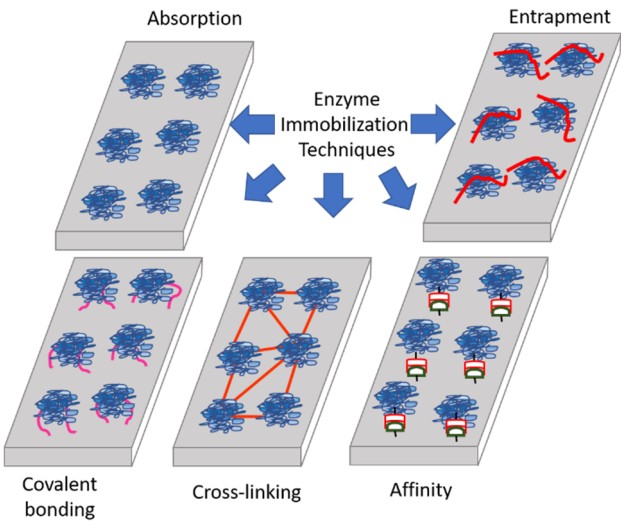

**Figure 11.** Various enzyme immobilization techniques for enzyme-based biofuel cells (EBFCs).

To obtain a summary of the progresses in the enzyme based biofuel cells over the last 30 years, the readers are referred to the works of Rasmussen et al. [261]. According to the authors, the biggest challenge in this field is the low stability and electrochemical performances of EBFCs. Babadi et al. [33] predicted that enzyme immobilization of EBFC electrodes can be greatly improved with novel nano carbon materials. For more details on EBFC-targeted carbon nano materials and functionalization, the readers are referred to their work [33]. Gonzalez-Solino et al. [262] summarized the most commonly used enzymes for EBFC anodes, wearable EBFC solutions, EBFC-based biosensors and provided a comparison of implantable EBFCs in terms of their output power. They hypothesize that these developments of cost-effective and safe EBFCs can power up key biomarker monitoring and help saving millions of lives. Gamella et al. [263] have focused on the interfacing technologies between EBFCs. They hypothesize that these cells, operating either internally or externally on a human body, could pave the way of bionic human-machine hybrid and this scope can extend to cyborg animals as well and can greatly contribute to environmental monitoring, homeland security and military applications. Jeerapan et al. [264] have summarized the key bottlenecks of wearable EBFC technology. They focus on the fact that the sustainable development of wearable EBFCs must be able to dynamically adjust with uncontrolled body changes that occur due to regular movements.

Carbon nanomaterials are extensively used in fabrication of enzyme-immobilized electrodes for EBFCs owing to the thin diameter, a feature that makes electrodes accessible to the enzyme active sites [33]. For example, Göbel et al. [265] reported their work where the anodes were fabricated with carbon nanotubes (CNTs) and modified with a polyaniline film and the cathode was made with PQQ (pyrroloquinoline quinone)-modified carbon nanotubes. Chung et al. [266] developed a carbon nano-flower structure which can be successfully utilized for immobilizing enzymes for EBFC application. Another study reported the use of spray coating for producing flexible biocathodes [267]. In this study, the target of effective enzyme wiring was achieved with flexible biocathodes. These cathodes were spray coated using a conductive ink composed of carbon nanotubes dispersion. Thin continuous layers of CNTs were successfully coated on to of a gas diffusion layer paper with a variable thickness 1 and 7.8 μm and were later with Laccase enzyme.

A very concerning feature for implantable components, and thus for EBFCs, is their host immune response. This is of utmost importance for any implants where living bodies react to unfamiliar materials through a series of reactions that would lead to the formation of a capsule of collagen around any external component. However, EBFCs require regular exchange of analytes through the cell and any protective collagen layer around the implantable device limits this mandatory transition of analytes [268]. Moreover, there are also possibilities of extracellular matrix infiltration inside the EBFC which can cause degradation of the immobilized enzymes [269]. A pictorial description of the concept is given in Figure 12.

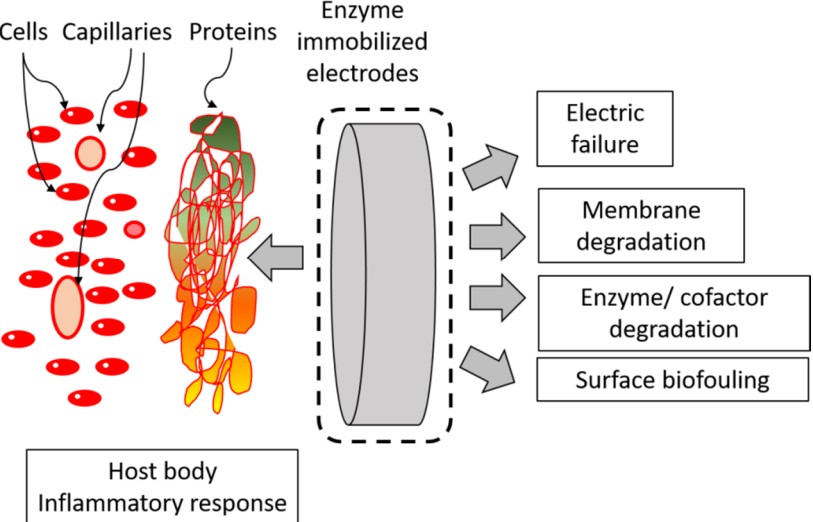

**Figure 12.** Common reasons for implanted EBFC failures.

### 4.2. Classification of EBFCs

The motivation for developing EBFCs was backed by the demand of electrical power requirements for monitoring [270] and maintaining [258] physiological parameters in living macro-organisms. So far, successful operations of EBFCs have been employed in plants [270–273], insects [9], mollusks [8], lobsters [34] and mammals [35]. We can classify EBFCs into four major groups: in vitro, plant-powered, animal-powered and human-powered. Animal-powered EBFCs can be further classified into externally and internally implantable sub-groups. Nearly all presently available human-powered EBFCs fall under the wearable sub-group. This classification is depicted in Figure 13.

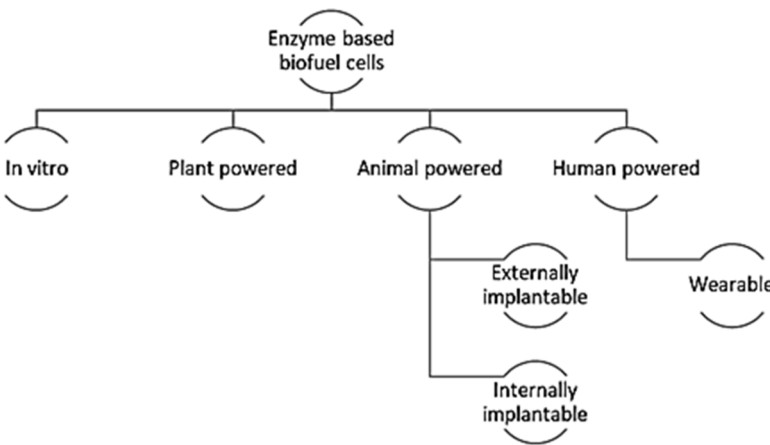

**Figure 13.** Classifying EBFCs according to their placement.

Studies on in vitro EBFCs, i.e., EBFCS which have been tested outside any living organism and in a lab environment, have been widely reported. One such experiment was presented by Castorena-Gonzalez et al. [274] where bucky paper electrodes were used. The anode was modified with PQQ-dependent glucose dehydrogenase, which is a quinoprotein enzyme with PQQ cofactor for facilitating glucose oxidation [275], and the cathode used laccase. The cell was initially tested in human serum solution and then on exposed rat cremaster tissue. Szczupak et al. [276] informed about an EBFC with a bienzymatic trehalose glucose oxidase trehalose anode and a bilirubin oxidase dioxygen cathode. Yin et al. [273] reported their study on a needle-type biofuel cell and tested it on artificial blood glucose and from the blood glucose in a mouse heart.

Initially and until now in the primary phase, the EBFCs have been tested in plant mediums. Flexer et al. [270] utilized cactus leaf and a light source to implement a biosensor for continuous monitoring of $O_2$ and glucose generation rate. MacVittie et al. [271] reported their enzyme-based bioreactor for orange pulp and the generated power was used to power a wireless data transmission system. A similar experiment using orange pulp for powering a wireless transmitter was reported by Holade et al. [272]. The authors worked on an EBFC in which electrodes were modified with inorganic nanoparticles deposited and with carbon black. Yin et al. [273] reported a needle type biofuel cell whose performance was tested in various fruits and 55, 44, and 33 μW of power output was obtained from specimens of grape, kiwifruit, and apple, respectively.

There is an interesting bio-hybrid EBFC implanted animal group that includes experiments where the EBFC (main cell) is placed outside the energy-harvesting organism. It is inspired by the observation that placement of a complete cell inside the receiving living organism becomes difficult for small-sized organisms like snails [8]. A similar configuration has also been used in live American lobsters [34]. A cyborg insect, *Blaberus discoidalis*, was reported by Schwefel et al. [9] where the insect hemolymph was used for the substrate for a trehalose/oxygen biofuel cell. As a continuation, a similar system was implemented in a moth for powering up a wireless sensor for environmental monitoring [12].

There are also examples of internally implanted EBFCs in animals. Cinquin et al. [277] reported the first in vivo implantation employed in the retroperitoneal space of moving rats. Zebda et al. [32] reported a similar configuration in rats, capable of producing an average open-circuit voltage of 0.57 V. Nadeau et al. [278] reported a cell implanted in the gastrointestinal tract of pigs that could perform in vivo temperature sensing and wireless communication. Ichi-Ribault et al. [35] reported implantation of another abdominal implant in a rabbit that operated for 2 months.

A new group of EBFCs are now being investigated which can be implanted externally. For example, Toit et al. reported power generation from a transdermal extract of pig skin [279].

Scopes of utilization for such external or wearable EBFCs for human use are also being explored. For example, studies on the use of an enzymatic biofuel cell on a contact lens [280–282] and patches [283,284] have been reported. Generation of electrical power from human perspiration are also reported [13,285].

Because of the present limitations in invasive cell implantation, research scopes took a turn towards the possibilities of using EBFCs with physiological fluids as an alternative to blood [262].

## 5. Research Progress in Biomechanical Energy-Harvesting Technologies

There has been an increasing demand for biocompatible and environment friendly alternative energy sources for next generation of low-power, wearable electronic components [1] and biomechanical energy-harvesting technology has emerged as a promising solution. They offer an eco-friendly and non-invasive power solution for monitoring, diagnostic and therapeutic operations. These devices utilize mechanical energy produced from living animals and convert them into power solutions.

### 5.1. Biomechanical Energy-Harvesting Mechanisms

According to Dong et al. [286], the four major bio-mechanical energy harvesters are piezoelectric, electromagnetic, electrostatic, and triboelectric mechanisms. In most cases, these mechanisms fall either within the vibration-based energy-harvesting technologies or motion-based energy-harvesting technologies.

Piezoelectric materials with cantilever geometry have been the classic choice [287] for mechanical energy harvesting. These materials are a subset of ferroelectric materials and produce an electrical charge when being mechanically deformed [288]. Deterre et al. [14] reported a micro-spiral piezoelectric energy harvester for extracting energy from regular blood pressure variation, Shafer et al. [289] reported an energy-harvesting system from the wings of flying birds and bats that could be used to power environmental monitoring systems. Vibration-based piezoelectric energy-harvesting mechanism is more applicable for insects or animals with flapping wings than from human joint movements. Currently with piezoelectric technology, it is possible to generate around 20 watts from foot strikes as well as from movement of the body's center of mass, 60 watts from ankle movements, 30 watts from both knee and heap movements, 2 watts from both elbow and shoulder movements [290]. A generalized piezoelectric-based energy-harvesting system is depicted in Figure 14.

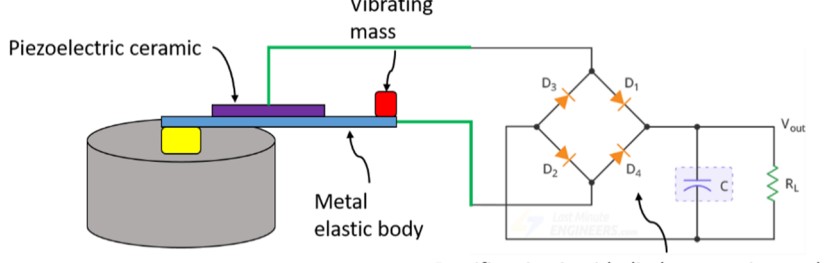

**Figure 14.** Piezoelectric based biomechanical electricity generation principle, redrawn from the schematic shown in [291].

An electromagnetic induction principle has also been utilized to extract biomechanical energy. Zurbuchen et al. [292] reported their prototype for harvesting energy from endocardial heart motion by electromagnetic coupling and the system has been tested in vivo in domestic pigs. The proposed energy-harvesting device consists of serially aligned copper coils, surrounded by a linear arrangement of permanent magnets are suspended between two spiral springs. The permanent magnet stack oscillates in the case of motion, which in this case is the heart's endocardial motion. Nakada et al. [293] developed an elec-

tromagnetic generator, suitable for inserting in the abdominal cavity of the birds and tested their performance in chickens and pheasants and obtained a power average of 0.47 mW. The objective of this work was to power up a biosensor to test the antigen-antibody reaction of avian influenza. Powering up the sensor was performed by a tiny generator with an electromagnetic induction coil implanted in the bird's abdominal cavity. The generator supplied power when chickens walk and pheasants flew. Almansouri et al. [294] efficiently used a magneto-acoustic resonator for tracking aquatic animals. The authors developed a system that converts low-frequency fish motion to excite high-frequency acoustic pulses. The prototype was able to generate an average acoustic sound of 55 dB sound pressure level at 1 m of distance with a resonant frequency of 15 kHz. Electrostatic energy harvesters transform energy from changing in capacitance according to Coulomb's law for two parallel plate capacitors [295]. Although some examples can be found [296,297], these components did not gain much popularity for biomechanical energy harvesting.

Triboelectric nano generators (TENG) were first introduced by Fan et al. [298] and rely on the principle of electric charge separation at the friction of two surfaces, different in terms of nano-scale roughness and thus creating an electric charge layer, by imposing variation in the capacitance electric energy that can be obtained from such small, flexible systems. This tribo-electrification is the working principle creating the natural phenomena of amber effect and lightning [299]. Combining the principle of tribo-electrification and electrostatic induction, TENG were developed and have gained huge popularity for harvesting biomechanical energy. A generalized TENG energy-harvesting system is depicted in Figure 15.

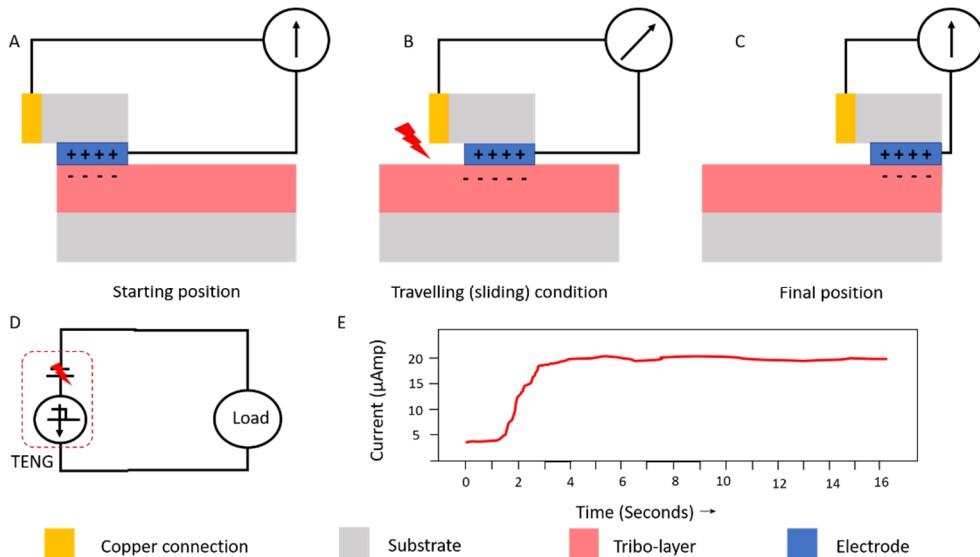

**Figure 15.** Triboelectric nano electricity generation (TENG) principle.

These nano generators, referred to as triboelectric nano generators (TENG) became very popular in a short time period, due to their flexibility [300] and cost-effectiveness. Zheng et al. [301] reported an in vivo biomechanical-energy harvesting using a TENG for the first time. Dong et al. [302] reported development of wearable, large-length, energy-harvesting textiles by incorporating TENG mechanism.

### 5.2. Energy Harvesting from Humans

Biomechanical energy-harvesting solutions gained much popularity thanks to their easily wearable features. Choi et al. [40] summarized the applicability of biomechanical energy harvesting from various form of human actions and motions including foot, knee, hip as well as upper limb motions and gave special focus to high power density, backpack like wearable rotary energy harvesters. Similar wearable harvesters have also been reported by Xie et al. [303], Yuan et al. [304] and Martin et al. [41] which generate watt level

power density. Gurusamy et al. [305] reported the identification of lower limb joints which would be better suitable for energy harvesting. Many research groups have concentrated on utilizing biomechanical energy from foot strikes through shoe soles [306–308]. Works are also found on biomechanical energy harvesting from ankle [309], knee [310–313] and hip [314] movements. A generalized wearable backpack based biomechanical energy-harvesting system is depicted in Figure 16.

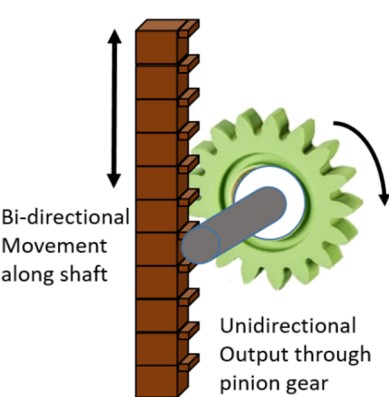

**Figure 16.** Rack-pinion gear based biomechanical electricity generation used in backpacks for utilizing up and down movements of the center of mass, redrawn from the schematic shown in [315].

Development of smart, energy-harvesting textiles with TENG components that would facilitate both electrical powering and wearable sensing applications has gained a considerable amount of interest over recent years. Example of such works include development of TENG-incorporated stainless steel/polyester fiber-blended yarns [302] and washable hybrid fibers with a piezoelectric-enhanced TENG mechanism [316]. Efforts have also been made to implement skin like transparent sheets that could harvest biomechanical energy. Relevant examples include, a textile-based TENG that can be actuated with skin touch [317]; a soft skin-like triboelectric nanogenerator that works both as an energy harvester and tactile sensor [318]; a transparent triboelectric, piezoelectric, pyroelectric combined hybrid nanogenerator with silver nanowires [319]; and an optically transparent, textile compatible TENG fiber with silk protein and silver nanowires [320].

### 5.3. Energy Harvesting from Non-Human Living Organisms

In addition to human-centered applications, biomechanical harvesters have also been tested for powering up cyborg-animal applications, targeted either toward environmental monitoring or observation of physiological parameters of the host animal. Aktakka et al. [42] reported a vibration energy scavenger for *Cotinis nitida* (Linnaeus) that could extract energy from the beetle's wing vibrations without affecting its movement. Zheng et al. [301] implanted a TENG-based energy harvester in a living rat for powering up a pacemaker. Nakada et al. [293] reported developing an electromagnetic generator implantation in the subcutaneous area or abdominal cavity of chicken-like birds and obtained a maximum 7 V peak-to-peak signal at 560/min of flapping of wings. Li et al. [321] presented an energy-harvesting module containing a flexible piezoelectric beam for extracting energy from fish movement to power up an acoustic transducer. Similarly, Almansouri et al. [294] utilized a "magneto-acoustic" resonator that converts low-frequency motions, ranging from 0.15 to 100 Hz into high-frequency acoustic signals. Shearwood et al. [322] reported a mountable energy harvester for bees that extracts energy from wing vibrations and powers up a radio frequency bee tracker with minimal physical hindrance.

### 6. Promising Bio-Energy Solutions, Lessons Learned

Our review of bio-energy harvesters focuses on three promising green energy resources. Tables 1–3 depict summaries of the key research components in the fields of

MFC, enzyme-based fuel cells and biomechanical energy harvesters, respectively, over the last decade.

**Table 1.** Summary of reviewed works on MFCs.

| MFC Focus Topic | Sub-Topic(s) | Studies |
|---|---|---|
| MFC Biofilm | | [53,68,77,159,217,227,229,323–327] |
| Dual chamber MFC | | [175,200,229,323,328–341] |
| Single chamber MFC | | [57,194,217,342–348] |
| MFC Anodes | | [20,49,61,63–65,349–372] |
| MFC Cathodes | | [73,75,76,90,99,106,107,116,161] |
| | Air cathodes | [63,65,70,74,77,126,344,348] |
| | Bio cathodes | [68,77,145,159,179,183,192,220,224,254] |
| | Algae/micro-algae bio cathodes | [72,177,193,195,196,217] |
| | Plant bio cathodes | [2,140,158,176,184,186–189,370,373] |
| Membranes | | [16,27,63,80,88,98,374,375] |
| | Cation exchange membranes | [85,86] |
| | Anion exchange membranes | [91–93,336] |
| | Porous, ceramic membranes | [87,95–97,373,376–379] |
| | Supported liquid ion membrane | [26,88,232] |
| PMS | | [23,62,109,211,380] |
| Remote power generation | | [6,11,23,24,61,65,68,107,110,123,127–129,135–137,140–142,151,172,197,199] |
| Waste processing | Waste-water processing | [19,29,58,60,67,73,114,121,132,149,155,194,202,203,216,315,322,325,328,330,336,359,360] |
| | constructed-wetland | [111,190,191,219,221–225] |
| | textile and dye processing | [140,217–220] |
| | solid waste processing | [242–246] |
| | metal recovery | [30,231–233,240,241,369] |
| Biosensing | | [11,28,31,111,129,143,154,194,201,202,247–255,354] |
| Powering robots | | [381–385] |

**Table 2.** Summary of reviewed works on enzyme-based fuel cells.

| EBFC Focus Topic | Sub-Topic(s) | Studies |
|---|---|---|
| Review articles | | [33,261–264,386–389] |
| EBFC cell components | Enzyme immobilization | [25,259,390,391] |
| | Anodes | [265–267,392–397] |
| | Micro-fluid structure | [398–402] |
| Tested with plants | | [270–273] |
| Tested with bio-hybrid organisms | Insects | [9,12] |
| | Molluscs | [8] |
| | Lobsters | [34] |
| | Mammals | [32,35,277,278] |
| Targeted towards external human use | Contact lens | [280–282] |
| | Skin patches | [13,279,285,403–405] |
| | Wearable fabric | [397,406] |
| Powering biosensors | | [12,33] |
| Powering organ on chip | | [407] |

**Table 3.** Summary of reviewed works on biomechanical energy harvesters.

| Biomechanical Energy Harvesting Focus Topic | Sub-Topic(s) | Studies |
|---|---|---|
| Review articles | data | [40,286,408,409] |
| Harvesting mechanism | Pezoelectrinc | [1,290,316,409–412] |
| | TENG | [1,36,298–300,302,307,316,319,320,408,413–415] |
| Harvesting from non humans | Insects | [42,322] |
| | Animals of higher order | [289,292–294,301,321,416] |
| From human wearables | Skin patches | [36,318,320,414,415,417] |
| | Smart textile | [316,317,413,418,419] |
| | Shoes | [306,411,412,415,420–424] |
| | Backpacks | [41,303,304] |

*6.1. MFCs*

MFC technology is the most promising bio-reactor with the possibilities of becoming the next green industrial alternative to conventional high carbon-footprint solutions [15,16]. However, the bottleneck remains at the point of finding suitable technologies for bringing the MFCs from the lab environment to the chemically harsh external world. For the application in wastewater treatments, the anodes must be robust for withstanding the impacts of diverse organic and inorganic components [60]. Presently, carbon based anodes are preferred for their bio-compatibility, porosity, corrosion resistance and conductive features. We predict that this research trend will continue for a few more years until an optimal anodic configuration is achieved.

For cathodes, two major concerns were prevention of bio-fouling [65] and ensuring continuous supply of electron acceptors [31] at cathodic chamber. In the field of MFC membranes, new porous, ceramic membranes have emerged as a promising solution and our prediction indicates that we shall find more economically viable and mechanically robust solutions from this line of research.

Moreover, some research groups consider MFC as the most suitable technology for providing energy autonomy to robots since the process can convert organic waste directly into electrical energy and this concept was proven by the 'Ecobot's [382,383] and 'Slug-bot' [381]. Ieropoulos et al. [425] and Mathuriya et al. [385] focused on this concept in their reviews. Thus, an interesting field is emerging where MFC will be utilized as a robotic gut to power next-generation, bio-inspired robots. Early generation of such robots were limited by their substrate extraction mechanism. At this point, the challenges are twofold: to design small-scale energy efficient MFC cells and to develop robotic systems for engulfing organic waste from their surrounding [384,426]. These examples indicate a next generation of MFC-powered energy autonomous robots, capable of sustaining by themselves without requiring additional intervention for power maintenance.

In order to bring this output to an applicable range, MFC-based fuel cells require efficient and dedicated power management systems. The reviews by Wang et al. [200] and Abavisani et al. [211] give a quick overview of the recent progress in the dedicated power management systems for MFCs where Wang et al. [200] elaborately summarizes MFC energy harvesting systems, methodologies and components Abavisani et al. [211] focuses more on the newly emerging maximum power point tracking (MPPT) systems. The major challenges in energy harvesting from MFCs for making them an alternative energy solution can be outlined as:

1. MFCs provide much lower volumetric power densities for bigger applications compared to smaller ones [147].
2. The power generation mechanism of MFC systems is not inherently self starting and usually require additional jumpstarting technology [200].
3. They do not offer the flexibility of stacking cells for increasing voltage and current ratings since a slight voltage mismatch creates local voltage reversal circuits and reduce the total output [205].

4.  These electricity power cells are actually living organisms and their dedicated power management systems need to correspond and adapt to biological activities of the cells and adjust with their continuously changing power curve [209].

### 6.2. Enzymatic Bio Fuel Cells

Since biofuel cells with abiotic electrocatalysts like platinum [427] or microbial [428] catalysts should not be implanted in living organisms for high inflammation risks, enzyme-immobilized EBFCs remained as the only viable option and many research groups have focused on this specific field. However, host immune response is still an important bottleneck [262] because the collagen capsule, generated as a natural response to an external implant, can limit the access of analytes [268] and initiate degradation of immobilized electrons [269]. We have summarized recent EBFC works based on their concentrations in Table 2.

At this point the major challenges for implementing EBFCs in the real world are:

1.  Insufficient output voltage level [429].
2.  Limited performance due to incomplete oxidation by the dedicated enzymes [259].
3.  Demand for an operating range of pH and temperature [259].
4.  Selection (unavailability) of optimal substrates inside a living body which will provide fuel for the cell, without causing immune response in the host and can accommodate with the bio-fouling conditions caused by the host body [262,387].
5.  Durability is the other bottleneck factor since lithium batteries for implanted devices can operate on average for five years [430], while the longest in vivo operating time for EBFCs is about two months [35].
6.  In terms of EBFC fabrication, the major challenge lies in effective enzyme wiring for efficient direct electron transfer mechanism [267].
7.  An additional concern is low oxygen concentration at cathodes which limits the performance of such biofuel cells.

Micro-fluid structures for EBFCs have also been widely studied since they offer advantages in terms of easier fluid manipulation, faster chemical response and higher surface to volume ratio [401]. From Table 2, it can also been seen that interest has shifted towards partial [280] or completely external [283] EBFC configurations in order to overcome the major bottleneck of internal host body reaction. Studies on the use of an enzymatic biofuel cell on a contact lens [280–282] and skin patches [283,284] support this hypothesis. For such externally wearable configurations, more concentration will be made towards developing base component for EBFCs which would be compact, flexible, biocompatible as well as mechanically and chemically stable [262]. Another alternative approach towards the solution of reducing host body immune system could be merging biodegradable and edible electronics [431] and targeted drug-delivery [432] technology with EBFCs which would stay in the host body system only for a short while and could be taken as regular medication. Application of EBFC fueled bio-energy harvesters in cyborg animals [34,258,433] are also found which show the promise of a new generation of convenient and energy autonomous environmental and agricultural monitors.

### 6.3. Biomechanical Energy Harvesters

The biomechanical energy harvesters are good candidates for powering up environment friendly, low-power, wearable electronic gadgets [1]. They are widely investigated for powering up external implants [304,314]. Researchers are also interested in utilizing such harvesters for powering up internal implants. However, these mechanical energy harvesters are definitely not intended towards large-scale or industrial power applications.

Easy wearability is considered as one of the main reasons behind the prominence of biomechanical energy harvesters. The recent works on biomechanical energy harvesters have been summarized in Table 3 according to their functionalities.

The Table infers that TENG energy is immensely popular these days and we believe the trend shall continue for the next decade. The use of smart clothes is on the rise and

our prediction is that most of these smart clothes will be powered by TENG energy. We assume that skin-like TENG components [318] will also be useful for immersive augmented reality applications. Portable and wearable energy harvesting modules like backpack energy harvesters [304] will be significantly helpful for powering up the next generation of implants as well as for exoskeletons for factory workers and army troops.

The major challenges for implementing efficient biomechanical energy harvesters can be summarized as:

1. Tuning resonant frequencies for vibration-based energy harvesting depending on various environments and situations [286].
2. Another key challenge for vibration-based energy harvesting is how to match frequency between the energy harvester and ambient environment to include a wider frequency bandwidth [286].

Variations of biomechanical energy harvesters, especially electromagnetic [292,293] and magneto acoustic types [294], have already been tested on animals. In this line of progress, we are hopeful that soon we will find a new generation of ethical, energetically autonomous and cyborg bio-hybrid animals, capable of eco-friendly monitoring of natural habitats at a large scale.

## 7. Conclusions

In our review, recent research progress on three promising bioenergy harvesters and their wide applicability is critically discussed. From these three technologies, MFCs show the promise of becoming a strong alternative to high carbon footprint solutions which is of utmost importance in today's world [1]. The potential of MFCs are limited by low power density, lack of internal robustness, difficulties in scalability and lack of portability. Nonetheless, the recent developments in photo-reaction coupled MFCs, improved electrode configurations, availability of robust ceramic membranes and novel technologies for substrate extraction show the promise of overcoming the limitations mentioned above.

While MFCs are perfectly suitable for industrial applications, they are not favored for clinical and bio-hybrid applications. However, in such fields, EBFCs and biomechanical energy harvesters are more convenient. Although EBFCs were initially considered for long-term internal implantation inside living bodies, the present trend is to make them either wearable or designed for short-term internal implantation. Biomechanical energy harvesters are more of external type and offers advantages in terms of wearability. Recent development in TENG has allowed this principle to be widely utilized in smart clothes and smart shoes as well as offered possibilities of augmenting tactile sensory perception with wearable skin-like configuration.

In addition to powering up human prosthetics and implants, both EBFC and biomechanical energy harvesters are also being intuitively used to power bio-hybrid organisms, which show new possibilities in the field of sustainable and natural environmental monitoring.

**Author Contributions:** Conceptualization A.S.A. and D.R.; methodology, A.S.A. and D.R.; literature review, A.S.A., D.R., F.I. and C.S.; writing—original draft preparation, A.S.A.; writing—review and editing, A.S.A., D.R., F.I. and C.S.; supervision, D.R. and C.S.; All authors have read and agreed to the published version of the manuscript.

**Funding:** This work was funded by the H2020 FETOPEN Project "Robocoenosis—ROBOts in cooperation with a bioCOENOSIS" (899520). Funders had no role in the study design, data collection and analysis, decision to publish, or preparation of the manuscript.

**Institutional Review Board Statement:** Not applicable.

**Informed Consent Statement:** Not applicable.

**Data Availability Statement:** Not applicable.

**Conflicts of Interest:** The authors declare no conflict of interest.

## Abbreviations

| | |
|---|---|
| MFC | Microbial Fuel Cell |
| PEM | Proton Exchange Membrane |
| CEM | Cation Exchange Membrane |
| AEM | Anion Exchange Membrane |
| ORR | Oxygen Reduction Reaction |
| PMS | Power Management System |
| EBFC | Enzyme based Bio Fuel Cells |
| TENG | Tribo Nano Electric Generator |

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
