# Peer review of "Towards Bio-Hybrid Energy Harvesting in the Real-World: Pushing the Boundaries of Technologies and Strategies Using Bio-Electrochemical and Bio-Mechanical Processes"

_applsci, doi:10.3390/app11052220_

Round 1
Reviewer 1 Report
This paper provides a comprehensive review of three bioenergy harvesters and their applications. The discussion of recent research progresses on these three technologies are well organized and thorough. I recommend to publish the paper.
Author Response
This paper provides a comprehensive review of three bioenergy harvesters and their applications. The discussion of recent research progresses on these three technologies are well organized and thorough. I recommend to publish the paper.:
We would like to thank reviewer for his/her availability to review our manuscript and the kind remarks.

Reviewer 2 Report
See comments in attached file.

Author Response
Reviewer #2:
In the manuscript “Towards Metabolism-Based Energy Harvesting in the Real-World: Recent Advances in Bio-Electrochemical Systems” the authors intend to describe the recent progresses on three different bio-energy harvesting systems including microbial fuel cells, enzyme based fuel cells and biomechanical energy harvesters. I consider that this review is of significant importance for the scientific community, however it requires major changes before it is accepted for publication::
We would like to thank reviewer for his/her availability to review our manuscript and guiding us thoroughly for improvement of the article. Below are the reviewer’s comments and their corresponding replies.
Outlined comments
- In order to make the document easier to follow the authors should choose a few words, and make a glossary with their description and their acronyms. Furthermore the authors should check the text to make sure that the acronyms are all correct. For example microbial fuel cells, are sometimes with its acronym MFC, and other times are still with the full name.
|
Glossary of the commonly used abbreviations
|
We agree your with have included the following terminologies in the glossary which is in page 2:
- Other example, is the acronyms BMFC and SMFC (line 316) used in the text that may confuse the reader. They should not be used, and these systems should always be referred Benthic MFC and Sediment MFC. It would be easier to read.
We agree with you have updated all the phrases involving BMFC and SMFC :
- Page 6, Line 189
- Page 10, Line 332
- Page 10, Line 337
- Page 10, Line 347
- Page 10, Line 348
- Page 10, Line 353
- Page 10, Line 354
- Page 11, Line 362
- Page 10, Line 369
- Page 10, Line 374
- Page 17, Line 661
- Page 10, Line 352
- Page 11, Line 356
- Page 11, Line 393
- Page 13, Line 461
- Page 17, Line 656
- Page 19, Line 765
- The acronyms EBFC and GBFC are also confusing in the text, and they should be only referred as EBFC – enzyme based biofuels cells. Indeed the authors describe these systems twice (line 653 and 661).
We agree with the reviewer and have updated all the phrases using ‘EBFC’ and removed duplicate terms from page 20.
- There are several acronyms in the text that are not necessary since they are just stated once or twice. These include TMFC, CNT, CW, MEC, DBMFC, PMU, PMFC, PAMFC, mMFC that are not necessary and should be removed from the text. They just confuse and mislead the reader.
We agree with the reviewer and have removed the acronyms in the following lines :
- TMFC –
- Page 12, Line 418
- CNT –
- Page 11, Line 367
- CW –
- Page 18, Line 684
- Page 18, Line 694
- MEC
- Page 19, Line 736
- Page 19, Line 744
- DBMFC
- Page 11, Line 375
- Page 16, Line 614
- PMU
- Page 16, Line 616
- PMFC
- Page 15, Line 562
- Page 15, Line 540
- PAMFC
- Page 15, Line 551
- mMFC
- Page 15, Line 574
- The section materials and methods does not make sense in this manuscript. This is a review, and for that reason there is no new data, only data available in published papers
We agree with the reviewer that this section name can be misleading since it is not a research article. However, we utilized a dedicated literature search methodology and consider this to be an important feature. Therefore we have changed the name into “Literature search method”.
- The title of the manuscript does not describe well the content of the review. The sentence “advances in Bio-Electrochemical Systems” should be removed, since the authors also focus on other bio-energy harvesting systems that are not bioelectrochemical based. Furthermore, the title include metabolism-based energy harvesting devices, and it is not clear how the biomechanical devices produce energy from the metabolism of living organisms.
We appreciate the reviewer for identifying this very important feature. Based on you suggestion we have changed the title to “Towards Bio-hybrid Energy Harvesting in the Real-World: Technologies and Strategies Pushing the Boundaries of Bio-electrochemical and Bio-mechanic processes”.
- The structure of the manuscript should be reviewed: There are some sub-sections that do not make sense taking in consideration the classification of the systems by the authors. For example, section 3.1 is refereed as MFC architecture, however the authors do not describe the different architectures of the MFC, but mainly how MFC works.
We agree with the reviewer and have reorganized the sub-section heading as follows:
“3.1 Principles of electricity generation with MFC”
- The organization of the sub-sections in the manuscript should also be organized better. For example section 3.1.1. Power management systems does not makes sense, and it is not clear what the authors wanted to describe here. This section should be in the applications of MFC, and not here. The authors should remove this section and organize the application of MFC section. Furthermore the challenges described in energy harvesting from MFCs do not provide any information regarding the challenges to make MFC a real-world application. This part of the section should be part of a different section – section 6 (see below).
We agree with the reviewer and performed the following changes:
- We have removed section 3.1.3 Power management systems
- Major part of the indicated section has been merged with sub-section 3.4.1 at page 16 Line 620:
“Lab-scale MFCs, which are in the order of milliliters in volume [102] generate electricity in the range of mW/m2 power density [11,103,104] and around mA/m2 in current density [105]. Up until now, most of the dedicated power management systems have been developed for marine remote sensors with low power requirements [106]. The popular choices for an MFC power management systems (PMS) are capacitors [109], charge pumps [23], boost [111] and flyback converters [112,113]. When using capacitors, there are two modes of operations: intermittent energy harvesting (IEH) [23] where energy is first stored and then released in bursts and continuous energy harvesting (CEH) mode [109] where a continuous output power is maintained at the load side. IEH mode has been proven to be more suitable for MFC based power generation systems [114]. The most simple form of an MFC PMS is an IEH mode capacitor circuit that charges one or more capacitors until enough energy is accumulated for powering up the load. While it’s operation is simple and straightforward, the output voltage is limited by the open circuit potential of the used capacitor [106].
Structurally, charge pumps are slightly more complex than capacitors and yet found at low cost and are widely used in MFC PMS [23], which is basically an inductor less DC/DC converter with capacitors. The S-882Z series charge pump from Seiko Instruments are very popular for MFC applications [106]. A DC/DC boost converter is a customized electric circuit that converts direct current power from one voltage level to a higher level [106]. Example of popular ones for MFC include ‘L6920DB from STMicroelectronics’ [106] and LTC3108 from ‘Texas Instruments’ [110,111]. Flyback converters [113] are more complex in the sense they use a coupled inductor instead. Theoretically their voltage gain can be infinite but are only applicable for below 100W power options [115].
A recent addition to MFC power management system development is application of maximum power point tracking (MPPT) technology [107]. Many other components including voltage balancing circuits [116], super-capacitors [11,28,117] and semiconductor devices [118] have also been used.
For more information regarding a summarized version of recent progresses on MFC based energy harvesting systems and their dedicated power management systems, the readers are referred to the reviews by Wang et al. [106]. For the recent progress on dedicated power management racking systems, please refer to the works of Abavisani et al. [107]. ”
- And the challenges have been included on section 6
- Furthermore in the classification of MFCs the authors classify MFCs regarding their placement, catalytic activities and applications (as described in Figure 4). However the names used in the sections do not match the names in the Figure. For example In situ MFC are divided in aquatic and terrestrial MFC, however section 3.2.2.1 describe sediment docked MFCs while section 3.2.2.2. describe floating MFC, that are both inside the aquatic MFC. Furthermore, the section 3.2.2.1 described as sediment docked MFCs, also contain information regarding benthic MFC. The differences between these two MFC should be explained better to not mislead the readers.
The sections have been reorganized into sub-sections ‘3.2.2.1. Aquatic MFCs’ and ‘3.2.2.2 Terrestrial MFCs’. From literature review we also found that both benthic and sediment MFCs indicate the similar configuration where the anode is buried in the sediment. For clarity we have added two more statements to clarify the ambiguity within sediment docked, benthic and sediment MFCs:
“Benthic MFC or sediment MFC are possibly the most studied configuration where bacterial metabolic activity occur at anaerobic, buried anodes and therefore them we refer in our classification as sediment docked MFCs [129].”
- Section 3.3.3 does not reflect the different organisms that can be used in MFC, since in this section, photosynthetic organisms are mainly used in cathodes, and the classification made by the authors are based on organisms that can be used in anodes. This section should be included inside the bacteria MFC section, since in most described situations bacteria are in the anode. Furthermore the title of this section should be Photoreactor aided, to reflect what is in Figure 4B.
We have upgraded the section name.
- In section 3.3.4, the authors describe impact of animal grazing of MFCs, but this is not described in Figure 4B. It is not clear the importance of this section and its impact in MFC.
We agree with the reviewer that it should not be within a sub-section and does not correspond with the figure. However, we wanted to inform the reader that animal grazing has impacts in real world MFC performance. So, we removed the sub section while rearranging the placement of the text in later part of section 3.3.
- Section 3.4.2 and 3.4.3 should be revised, since there are some work regarding bioremediation, and it should be included in a different section. Considering the application of MFC in removal of contaminants a bioremediation section should be included in the MFC applications section. Indeed line 627-632 describe work performed in metal recovery, which is not relevant for the solid-waste processing, but it is for bioremediation. Furthermore the title solid-waste processing in not described in Figure 4C as one of the main applications, and should be included in the wastewater treatment. The bioremediation section/metal recovery, should also be included in this section.
We thank the reviewer for giving us a detailed suggestion. In order to incorporate them we have made the following changes in the manuscript:
- We have upgraded the application based classification of MFCs in Figure 9
- We have removed the suggested contents from the solid waste processing sub section and re-organized their placement into the subsections wastewater processing and bioremediation.
- We also have introduced the following section focusing on the applications of bioremediation by MFCs
“3.4.3 Bioremediation
An additional application of MFC technology has been bioremediation and also one of the most investigated applications of sediment MFCs [233]. In this process microorganisms are utilized to treat polluted to sites to break down environmental pollutants, to regain back their original condition [234]. This has long remained as an alternative natural process of waste removal from land [235]. For more details on recent developments on bioremediation of sediments, please refer to the works of [236]. There have been particular in removing in removing organic [237] hydrocarbon [238] and metal [239] based pollutants .
In their review, Dominguez-Benetton et al. [240] summarized the latest mechanisms of metal recovery using microbial fuel cells. In another review, Wang et al. [229] classified mechanisms of metal recovery using MFCs in 4 different categories: direct metal recovery with abiotic cathodes; metal recovery with externally powered abiotic cathodes, metal conversion with bio-cathodes and metal conversion with externally powered bio-cathodes supplemented by external power sources.”
- Section 4.1 is entitled as Architecture of EBFC, however there is no description on the difference architecture of these systems. This should be included here, or the title of this section should be modified. This section should be more clearer to the reader. The author mainly describe what is the literature and the differences between EBFC are less explored. Furthermore classification of EBFCs should be described in section 4.2, however no classification is stated.
- We have modified the section name to “Principles of electricity generation from enzyme based biofuel cells”.
- Included additional text at Line 836 and figure (Figure 10):
“EBFCs are bio-electrochemical cells that can extract energy from glucose and alcohol based organic substances [259]. Although their principle of operation is similar to MFCs, both anode and cathode of such bio-fuel cells are prepared by embedding pure enzymes [259].”
- We have added the following section at Line 903 to complete the classification:
“We can classify EBFCs into four major groups: In-vitro, plant powered, animal powered and human powered. Animal power EBFCs can be further classified into externally and internally implantable sub-groups. Nearly all presently available human powered EBFCs fall under the wearable sub-group.”
- In section 5 four biomechanical energy harvesters devices are described, however only a small description is made on them. Given that this review is focused on energy harvested devices, more information about these devices should be included in the review, mainly how they work and how mechanical energy is converted in power solutions. Furthermore, the different configurations and applications should also be provided. A similar organization of the sub-sections as it was described for MFC, should be provided for these systems. This section should be revised, since the authors only describe a list of possible devices but do not describe how these systems work and the impact of the biological processes, given the focus of the review in metabolism based energy harvesting devices.
We agree with the reviewer that this section required more explanation on the biomechanical energy harvesting mechanism. Since electrostatic based energy harvesting is least utilized, we added two more schematics explaining piezoelectric based energy harvesting mechanism and the principles of backpack based, electromagnetism driven energy harvesting mechanism along with previously described TENG mechanism. However, we are concerned that increasing more sub-sections will reduce the readability for the readers. So, please accept our decision not to include further sub-sections in this case.
“Figure 14: piezoelectric based biomechanical electricity generation principle, redrawn from the schematic shown in [296]”
“Figure 16: Rack-pinion gear based biomechanical electricity generation used in backpacks, redrawn from the schematic shown in [320]”
- Throughout the text the authors have sentences that describe what is in the literature in a way that is descriptive and not explanatory. For example in Line 236-240 the authors describe other reviews, which does not makes sense, taking in consideration that this is a review. The authors in this case can state that “for more information regarding specific subject see the review XX”.
We have updated the identified statements as followed:
“For more information regarding a summarized version of recent progresses on MFC based energy harvesting systems and their dedicated power management systems, the readers are referred to the reviews by Wang et al. [106]. For the recent progress on dedicated power management racking systems, please refer to the works of Abavisani et al. [107].”
- Again in Line 392-402 the authors are just listing what is in the literature without explaining the differences between the works, and their impact in the development of Terrestrial MFC.
We have updated the section on terrestrial MFCs according to the following:
“Efforts have also been given to develop MFCs that can operate outside waterbodies and are often referred to as terrestrial MFCs or TMFCs [160] where the cathode was air facing, and the anode burying soil acted as both electrolyte and proton exchange membrane. Huang et al. [161] reported their work where they used an insertion-type microbial fuel cell for remediation of phenol in waterlogged soil and electricity generation. Pietrelli et al. [2] reported use of terrestrial MFC to operate a wireless sensor network for land monitoring and precision agriculture where no water-soil interface was present and the anodes were buried at 8 cm depth. A similar application was reported by Adekunle et al. [154] reported the application of terrestrial MFC into a transportable bio-battery where they used a solid anolyte soil in addition to a small water reservoir for keeping the anodes functional. Simeon et al. [162] reported a soil-based, single chamber MFC with carbon felt electrodes where performance of the substrate (soil) was improved after urine treatment.
Another promising and emerging concept is integration of terrestrial MFCs with green infrastructures for achieving cleaner environment in urban landscape[163,164]. The green roofs can be of particular interest [164–166] for this purpose since in addition to electricity generation, they also provide cooling impact and thus reduce additional energy demand. An additional smart agricultural application should also be noted where MFC integrated power generation systems are utilized for plant heath monitoring applications[167,168].”
- In Line 434-449 the authors should not only list the work that have been done in Yeast aided MFC, but also describe their differences and their impact in this type of systems.
The following edits were performed to incorporate the reviewer’s suggestion:
“For more details on current state-of-the-art MFC systems that utilize fungi for biocatalysts please refer to the works of Sekrecka-Belniak & Toczyłowska-Mamińska [179] also summarized the current state-of-the-art MFC systems that utilize fungi for biocatalysts. Morant et al. [180] reported their primary experimental evidence that fungi isolated from the Caatinga region in Brazil can be used as efficient biocatalysts for MFCs. For their study they tested three Brazilian filamentous fungi Rhizopus sp. (SIS-31), Aspergillus sp. (SIS-18) and Penicillium sp. (SIS-21) and proved all three of them compatible for MFC air-cathode configuration. Kaneshiro et al. [46] reported examining MFC power using seven kinds of yeasts with a milliliter scale, dual-chamber MFC cell configuration with carbon fiber bundles as electrodes. They found Kluyveromyces marxianus as good candidate for biocatalytic activity and this finding can be particularly advantageous for power extraction from woody biomass. Mardiana et al. [181] reported use of yeast based MFC system with mediator methylene blue and electron acceptor K3Fe(CN)6. Impact of two inorganic mediators, employing two different mediators, methylene blue and methyl red were tested on Fungi-MFC was reported by Christwardana et al. [49] The authors found that yeast Saccharomyces cerevisiae performs better with methylene blue. Pontié et al. [182] reported a fungus Scedosporium dehoogii based MFC that can decompose acetaminophen, a widely used component in pharmaceutical industries whose degradation is quite difficult, oxidation in the anode and degradation of its main by-product. Along with decomposing acetaminophen , the fungal microbial fuel cell also provided a stable output of 50 mW/m2 .”
- In lines 565-573 and 580-599 the authors again describe the different manuscripts existent in the literature. They should describe their differences, and the impact in MFC applications. Lines 604-614 as well as 789-798 should also be revised.
In order to describe the differences better, we have updated the following sections:
- “Jia et al. [139] reported their works on power generation with single chamber MFC with brush anodes and carbon cloth based cathodes while processing food waste and found that different order of organic waste loading affects the MFC performance. Tee et al. [140] reported an MFC-adsorption hybrid system with air-cathode and with granulated activated carbon based anodes. For a more detailed review of MFC application on wastewater processing, please see the review study by Gude [215] and for impact od wastewater substrate composition, please see the works of Pandey et al. [216]. For going more into details of how to integrate wastewater processing with MFC technologies, please see the works of He et al. [60]. Do et al. [29] classified conventional wastewater treatment-MFC power generation systems into 5 major groups including sediment MFCs (SMFCs), constructed wetland MFCs (CW-MFCs), membrane bioreactor MFCs (MBR-MFCs), desalination MFCs (DS-MFCs) and others. The authors also performed comparison between the classes in terms of their substrate, power density, and chemical oxygen demand rate.”
- “Constructed wetland (CW) is a man-made wetland used for organic degradation for the wide range of agricultural to industrial wastewater [222]. Various efforts can be found on integrating the organic process of constructed wetland with MFCs. Oon et al. [222] built an up-flow constructed wetland where anaerobic and aerobic regions were naturally developed in the lower and upper bed and the system obtained a 100% chemical oxygen demand removal efficiency. Yadav et al. [223] reported use of another vertical flow constructed wetland system to remove different dye from synthetic wastewater and generate electricity as well. The configuration achieved the maximum value of 93.15% dye removal efficiency following a 96 h of treatment from the wastewater with 500 mg l−1 initial dye concentration. Villaseñor et al. [224] tested applicability of horizontal subsurface flow constructed wetland for performing simultaneous organic waste processing and power generation and found that such configurations can only handle low organic loading rate. This study offered two major observations: (a) and (b) the photosynthetic activity of the macrophytes, Phragmites australis was affected on the light/darkness changes caused voltage fluctuations and affected stable performance of the system. Liu et al. [225] demonstrated that use of Ipomoea aquatica plant in their constructed wetland MFC system provided a higher power density and nitrogen removal in comparison to their contemporary systems. The authors also worked on optimizing the vertical constructed wetland with 3 different electrode materials (stainless steel mesh, carbon cloth, granular activated carbon) and found both stainless steel mesh and granular activated carbon’s suitability for such configurations. Fang et al. [219] reported another successful combination of Ipomoea aquatica plantation constructed wetland–MFC system for azo dye decolorization. The planted system achieved the decolorization rate of maximum 91.24% with a voltage output of about 610 mV. Additionally, the system promoted growth of Geobacter sulfurreducens and Beta Proteobacteria while inhibited Archaea growth in anode. Srivastava et al. [226] found from their study that a coupled constructed wetland–MFC system performs better in removing organic substances than normal constructed wetland For a more detailed review of coupled constructed wetland and MFC system, please see the review by Doherty et al. [200,227]. In [200], the authors particularly stressed on the the importance of maintaining anaerobic anode and oxygenated cathode configuration separated. Corbella & Puigagut [227] indicated that constructed wetland systems naturally offer aerobic conditions in the upper layers and anaerobic in the deeper ones and results in favorable environment for MFC power generation system implementation. Xu et al. [201] identified high internal resistance as one of the limiting factors of such coupled systems. and offered a capacitor engaged duty cycling for solution. The researchers tested a new strategy, called capacitator engaged duty cycling (CDC), with an open air bio-cathode constructed wetland MFC system and obtained 19.81% more electric charge than the conventional continuous loading system.”
- “Mohan et al. [243] reported a solid phase microbial fuel cell system, developed to evaluate the potential of bioelectricity production by fermentation of food waste that gave promising results. The configuration utilized open air cathode sediment MFC configuration with graphite electrodes. They identified distance between the electrodes and PEM had a significant influence on power output and amendment of sodium carbonate improved system’s power buffering capacity. Lee et al. [244] reported another MFC based system for handling solid wastes as a feedstock. The authors evaluated the system with two configurations: (1) a single chamber combined membrane-electrode configuration; and (2) a dual chamber, proton-membrane-less configuration brush-type anode and double air cathode and used cow manure for feedstock and the second configuration provided better results with higher power output. Wang et al. [245] also reported a solid state MFC for processing cow manure with a single-chamber, air-cathode MFC configuration. The authors reported that a moisture content higher than 80% was suitable for current generation. Moreover, an addition of small amount of platinum catalyst improved the power density by 10-fold and output voltage by twice as much. Damiano et al. [246] reported their study of feasibility analysis of two MFC based electricity generation configurations that could simultaneously treat municipal solid waste landfill leachate. They identified that for such cases, smaller configurations perform better in power generation. Pendyala et al. [247] tested the practicality of using solid municipality waste as substrate for an MFC based system where they categorized the organic waste components in 3 main groups including food waste , paper–cardboard waste and garden waste and concluded that organic fraction of municipal solid waste is a promising feedstock for MFCs based waste processing. Detail data analysis from their observation indicate that the microbial composition of the anodic biofilm became a function of the feed composition. Moreover, they also found that regulating the protein content and removing furfurals and phenolic compounds from feedstock could increase the percentage of chemical oxygen demand removal rate.”
- “Electromagnetic induction principle has also been utilized to extract biomechanical energy. Zurbuchen et al. [293] reported their prototype for harvesting energy from endocardial heart motion by electromagnetic coupling and the system has been tested in vivo in domestic pigs. The proposed energy harvesting device consists of serially aligned copper coils, surrounded by a linear arrangement of permanent magnets are suspended between two spiral springs. The permanent magnet stack oscillates in presence of motion, which in this case is heart’s endocardial motion. Nakada et al. [294] developed an electromagnetic generator, suitable for inserting in the abdominal cavity of the birds and tested their performance in chickens and pheasants and obtained a power average of 0.47mW. The objective of this work was to power up a biosensor to test the antigen-antibody reaction of avian influenza. Powering up of the sensor was performed by a tiny generator with electromagnetic induction coil implanted in the bird’s abdominal cavity. The generator supplied power when chickens walk and pheasants flew. Almansouri et al. [295] efficiently used a magneto-acoustic resonator for tracking aquatic animals. The authors developed a system that converts low-frequency fish motions to excite high-frequency acoustic pulses. The prototype was able to generate an average acoustic sound of 55 dB sound pressure level at 1 m of distance with a resonant frequency of 15 kHz. Electrostatic energy harvesters transform energy from changing in capacitance according to Coulomb’s law for two parallel plate capacitors [296]. Though some examples are found [297,298], these components didn’t gain much popularity for biomechanical energy harvesting. ”
- Again in Lines 682-702 the authors describe other reviews instead of reviewing the topic. This should be modified.
We have updated the following section accordingly:
“For obtaining a summary of the progresses in the enzyme based biofuel cells over the last 30 years, the readers are referred to the works of Rasmussen et al. [262]. According to the authors, the biggest challenge in this field till now is low stability and electrochemical performances of EBFCs. Babadi et al. [33] predicted that enzyme immobilization of EBFC electrodes can be greatly improved with novel nano carbon materials. For more details on EBFC targeted carbon nano materials and the functionalization, the readers are referred to their work [33]. Gonzalez-Solino et al. [263] summarized most commonly used enzymes for EBFC anodes, wearable EBFC solutions, EBFC based biosensors and provided a comparison of implantable EBFCs in terms of their output power. They predict that development of cost effective and safe EBFCs can power up key biomarker monitoring and help saving millions of lives. Gamella et al. [264] have focused on the interfacing technologies between EBFCs. They hypothesize these cells, operating either internally or externally on a human body could pave the way of bionic human-machine hybrid and this scope can extend to cyborg animals as welland can greatly contribute in environmental monitoring, homeland security and military applications. Jeerapan et al. [265] have summarized the key bottlenecks of wearable EBFC technology. They focus on the fact that sustainable development of wearable EBFCs must be able to dynamically adjust with uncontrolled body changes that occur due to regular movements.
Carbon nanomaterials are extensively used in fabrication of enzyme immobilized electrodes for EBFCs owing to the thin diameter, a feature that makes electrodes accessible to the enzyme active sites [33]. In their review, Babadi et al. [33] highlighted implantable biofuel cells, focusing on the nano-carbon functionalization. Multiple studies have reported using of modified carbon nano-tubes CNTs for efficient fabrication of biofuel electrodes. For example, Göbel et al. [266] reported their work where the anodes were fabricated with carbon nanotubes and modified with a polyaniline film and the cathode was made with a PQQ (pyrroloquinoline quinone) -modified carbon nanotubes. Chung et al. [267] developed a carbon nano-flower structure which can be successfully utilized for immobilizing enzymes for EBFC application. Another study reports use of spray coating for producing flexible biocathodes [268]. In this study, the target of effective enzyme wiring was achieved with flexible biocathodes. These cathodes were spray coated of a conductive ink composed of carbon nanotubes dispersion. Thin continuous layers of carbon nanotubes (CNTs) was successfully coated on to of a gas diffusion layer paper with a variable thickness 1 and 7.8 μm and were later with Laccase enzyme.”
- Section 6 is not clear. It seems that the authors are making a review of the review. This section should be deleted and if necessary the authors should make a section on the current challenges of the different harvested devices. With this, several parts of the text present in other sections should be put here, including parts of section 3.1.1, lines 320-326 of section 3.2.2.1. and section 4.3.
According to your suggestion, we have eliminated a large section from section 6.1 and included challenges for real world application in sections 6.1, 6.2 and 6.3:
- “In order to bring this output to an applicable range MFC based fuel cells require efficient and dedicated power management systems. The reviews by Wang et al. [107] and Abavisani et al. [115] give a quick overview of the recent progress in the dedicated power management systems for MFCs where Wang et al. [107] elaborately summarizes MFC energy harvesting systems, methodologies and components Abavisani et al. [115] focuses more on the newly emerging maximum power point tracking (MPPT) systems. The major challenges in energy harvesting from MFCs for making them an alternative energy solution can be outlined as –
- MFCs provide much lower volumetric power densities for bigger ones compared to the smaller ones [160].
- The power generation mechanism of MFC systems is not inherently self starting and usually require additional jumpstarting technology [107].
- Does not offer the flexibility of stacking cells for increasing voltage and current ratings since a slight voltage mismatch creates local voltage reversal circuits and reduce the total output [108].
- These electricity power cells are actually living organisms and their dedicated power management systems need to correspond and adapt to biological activities of the cells and adjust with their continuously changing power curve [113].”
- “At this point the major challenges for implementing EBFCs in real world are –
- Insufficient output voltage level [288].
- Limited performance due to incomplete oxidation by the dedicated enzymes [261].
- Demand for an operating range of pH and temperature [261].
- Selection (unavailability) of optimal substrates inside a living body which will provide fuel for the cell, without causing immune response in the host and can accommodate with the bio-fouling conditions caused by the host body [264,289].
- Durability is the other bottleneck factor since lithium batteries for implanted devices can operate on average for five years [290], while the longest in-vivo operating time for EBFCs is about two months [35].
- In terms of EBFC fabrication, the major challenge lies in effective enzyme wiring for efficient direct electron transfer mechanism [269].
- An additional concern is low oxygen concentration at cathodes which limits the performance of such biofuel cells.”
- “The major challenges for implementing efficient biomechanical enrgy harvesters can be summarized as –
- Tuning resonant frequencies for vibration based energy harvesting depending on various environments and situations [291].
- Another key challenge for vibration based energy harvesting is how to match frequency between the energy harvester and ambient environment to include a wider frequency bandwidth [291].”
- The figures of the manuscript should be reviewed. To make them clear and have a more impact, some of them should be split and put it in the right place of the manuscript: The panels C, D and E of Figure 2 are not stated in the text and should be removed. In panel A of Figure 2, the flow of electrons should be clearer. In Panel B in Figure 2, the authors describe short and long distance electron transfer, which is incorrect in this figure. In these pathways there is always long-distance electron transfer since the electron donor and electron acceptor are in different cellular compartments. Furthermore, here the authors just want to describe the fact that is direct electron transfer with conductive pili and cell-surface proteins or indirect electron transfer in the presence of electron shuttles/mediators. The figure should be modified to contemplate the direct and indirect electron transfer. They should also state what are the small circles at the cell surface of bacteria.
Figure 2 has been split into different relevant parts. The electron paths in new Figure 2 are made bolder and Figure 3 has been updated for long distance direct electron transfer mechanism.
“Figure 3: Electron transfer mechanisms, mediators and biofilm for MFC technology.”
- In Figure 4 the authors describe the classification of the main components of MFC. Therefore it should be MFC Cathodes and not only cathode. Did not understand what is ‘cermaic membranes’ in MFC membranes. Figure 4 should be split into 3 different figures, that should appear in the text in the different sections (Figure 4A in section 3.2, Figure 4B in section 3.3 and Figure 4C in section 3.4.
We have split the figures according to your suggestions:
- Figure 3 has been split into 3 parts: MFC Anodes (Figure 4), MFC Cathodes (Figure 5) and MFC Membranes (Figure 6).
- Figure 4 has been splt into 3 parts: In section 3.3, section 3.3 and in section 3.4.
- We also have corrected the typo of the current figure to “Ceramic” (Figure 6).
- In Section 3.3.1. more information should be provided regarding bacteria that can be used in MFC. Most of the knowledge on MFC and their applications came from studies using bacteria. More information regarding these organisms, including the most used, and the mechanisms by which they transfer electrons to electrodes in MFC should be included.
Indeed, most of the knowledge on MFC and their applications come from bacterial studies. To incorporate the editor’s suggestions, we have included the following text in section 3.3.1::
“Two most studied exoelectrogen bacteria for MFCs are Geobacter sulfurreducens [54,172] and Shewanella oneidensis MR-1 [54,173,174]. Geobacter sulfurreducens only performs direct electron transfer mechanism, either through their extracellular pilin based cytochromes or the cell body itself [172]. The Shewanella species can perform both indirect electron transfer and direct electron transfer mechanism within MFC. For indirect electron transfer they utilize self secreted electron mediators and direct electron transfer is performed via outer membrane cytochrome c and nanowire [174].”
- In Figure 4B the authors describe other exoelectrogen based MFC systems. What are these, and why they are not described in the text?
Archaea based fall within this group of other exoelectrogen based MFC systems. We wanted to provide a glimpse that such MFCs also exist and included an additional statement in section 3.3 to clarify our objectives:
“Here we have classified MFCs based on their anodic biocatalysts into the following three major classes (bacteria based, yeast based and Archaea based MFC) as depicted in Figure 8 and this has been abridged from the classification exoelectrogens in [54]. In the review we focused on bacteria based MFCs and give a short overview of yeast based MFCs.”
- In section 4.2 a scheme as exists for MFC classification should be included. This would make this section clearer. Furthermore a real classification of these systems is missing, and it should be included here.
We have included the following text and the schematic:
“We can classify EBFCs into four major groups: In-vitro, plant powered, animal powered and human powered. Animal power EBFCs can be further classified into externally and internally implantable sub-groups. Nearly all presently available human powered EBFCs fall under the wearable sub-group. This classification is depicted in Figure X.”
- Line 81-82 – The sentence “In addition, fungi based MFC have also been reported” should be removed and the references added in the previous sentence.
We have incorporated the suggestion accordingly.
- Line 95 and Line 161 – it is not an external load resistance or external load circuit, but instead an external circuit. The external load resistance is only used to measure the current produced, but the electron flow occurs through an external circuit. Also external load circuit consumes electric power, and here it is only needed to transfer the electrons from the anode to the cathode.
We have updated the term in both the cases.
- Line 99-100 – this sentence does not makes sense. These three considerations have a significant impact in MFC performance, but not on MFC architecture.
We agree with you and have incorporated your suggestions:
“Bacterial electron transfer mechanisms and their dependence on mediators and biofilms are three major considerations of MFC performance.”
- Line 113 and Line 410– the authors describe that five classes of Proteobacteria are used in MFC, and that two major classes exists. However this may mislead the reader. Until know only members of these 5 classes of Proteobacteria were identified as electroactive, and that does not mean that other organisms from other class could not be electroactive. In order to not mislead the reader, the way that the authors describe this should be modified. Furthermore in line 410 it is described that the “5 classes are organized in two major classes”. Once again this may mislead the reader, regarding the classification of electroactive organisms. This sentence should be rewritten to not mislead the reader.
We rephrased both the sections for better clarification
- Line 122: “Microorganisms that can effectively generate electricity in MFCs without additional mediators include a few classes of Proteobacteria in addition to some microalgae, yeast, and fungi species [53].”
- Line 447: “For example, [54] classified use of electro-active microorganisms on MFC electrodes into two major classes: bacteria, archaea, eukaryotes on anodes and bacteria, archaea on cathodes. Again, these two major classes are further divided into five classes. While Kumar et al. [53] classified microorganisms for external mediator-less configuration that included five classes of Proteobacteria, identified till date, in addition to some microalgae, yeast, and fungi.”
- Line 114, 424 – replace exocellular with extracellular
In addition to two prior IJAMT reference, presently referenced as [34] and [35], we have added two more relevant references on residual stress from IJAMT.
- Line 114 – besides ref 53, this sentence should also include reference 158 and reference https://doi.org/10.1002/celc.201600079
We have included the reference suggestions accordingly.
- Line 118-120 – this sentence is not correct. One of the first organisms to show to produce conductive nanowires was Geobacter sulfurreducens. These nanowires were latter described to be conductive pili composed by the pilA protein. In the case of Shewanella, that also produced nanowires in specific conditions, the nanowires are extensions of their outer-membrane. Shewanella as well as Synechocystis do not contain conductive pilus-like appendages. Regarding Synechocystis they contain pili appendages but they have not been studied regarding their conductivity. For these reasons this sentence should be modified. The authors also mention Synechocyctis but do not provide a reference for this organisms, such as doi: 10.3389/fpls.2020.00241. The names of the organisms should also be italic. The authors should check all the document regarding microorganisms names.
We thank the reviewer for identifying this error. With your permission we would like to use part of your statement as follows and remove the previous sentence:
“A sub group of these bacteria are nanowire generators. Geobacter sulfurreducens was one of the first organisms to produce conductive nanowires. These nanowires were latter referred to conductive pili because of their composition with the pila protein. Bacterium Shewanella oneidensis can also generate electrical nanowires [54,55] under special condition and as extensions of their outer-membrane and transfer electrons to anode without requiring soluble electron shuttles.”
- Line 125 – the authors state that the biofilm is highly advantageous in MFC. However this is only correct for biofilms made by electroactive bacteria. If any other bacteria that are not conductive make a biofilm near the electrode, there would limit the access to the electrode. I recommend the authors to substitute “is” by ”can be”.
We have updated the sentence accordingly.
- Line 147 – The authors describe that stainless steel was found to be the only anode for MFC. However this is not correct, since depending on the applications carbon cloth electrode allows the production of higher current, as described in the literature. I suggest the authors to remove “only” from this sentence.
We have updated the sentence accordingly.
- Line 165 – the sentence “does not yet exist” should be removed. It is not clear what the authors meant.
We have rephrased the statement as follows:
“Ideally, the MFC cathode should be very reactive, capable of supporting ORR catalysts as well as remain low cost [66]. However, such an optimal MFC cathode configuration has yet not been achieved.”
- Line 166 – Insert “the” in the sentence “In the first generation of MFCs,…”
We have updated the sentence accordingly.
- Line 167-168 – the authors describe the poisonous effect of platinum and copper in the cathode of MFC, however how this may occur if in the cathode of MFC there is no microorganisms.
We thank the reviewer for identifying this important statement. It has been found that due to movement of micro-organisms even in presence of membrane, that catalytic activity of platinum and other ORR catalysts reduce after about a week and this is referred to as bio-poisoning of platinum. We included this explanation in the indicated section as follows:
“reduced capacity due to bio-poisoning caused by microorganisms [16] that sometimes can occur even in presence of the membrane [67,68]”
- Line 168-169 – The authors describe that oxygen or nitrates should be in the vicinity of cathodes. This is not clear when most of the cathodes use oxygen. It is not clear the presence of nitrates here. This section should be modified to make it clear.
It is true that in most cases the cathodes use oxygen. Therefore we removed the term nitrates in the mentioned statement and added an additional sentence afterward for clarifying our viewpoint:
“However, it has also been reported that in deep water column MFC configurations, there may appear an anoxic environment and in such cases, the ORR reaction is completed by other electron acceptors like nitrates, sulfates or iron oxides [72].”
- Line 180-184 – this sentence is too big and it is difficult to follow. I suggest the authors to split the sentence in two.
The sentence has been split and rephrased as follows:
“Air cathodes [70] have emerged as popular low cost solution providing sustainable aeration at cathodes. In this configuration the cathode surface partially remains open to air and continuously receives oxygen supply. Janicek et al. [76] reported a generalized air cathode configuration which consists of a catalyst layer that faces solution side of the cathode, a gas diffusion layer that faces air, and a conductive base material layer. The conductive layer also acts as a current collector as well as a mechanical support provider.”
- Line 186-188 – this sentence is a repetition and not necessary here. Therefore it should be removed.
We have removed the sentence.
- Line 203 – remove “the proton exchange membrane”, since depending on the system, different membranes can be used.
We have updated the sentence accordingly.
- Line 204-2011 – since this is a list, each sentence should end with a comma.
We have incorporated the suggestion accordingly.
- Line 253 – PMS is not described yet in the text
We now have introduced the term at this point:
“The popular choices for an MFC power management systems (PMS)…”
- Line 258 – It is not clear what the authors mean by MFC bio-reactor. Aren’t all MFC a bio-reactor?
Yes, all MFC systems are bio-reactors. In order to remove the ambiguity we have rephrased the statement as follows:
“IEH mode has been proven to be more suitable for MFC based power generation systems [110].”
- Line 332 – The classification of BMFC into open- and closed-water BMFC was proposed by the authors, or by others? It is not clear where these systems differ, and why the different nomenclatures. Furthermore the differences between open and closed water can also be applied to sediment MFC. This should be made clear.
The classification between the open and closed water benthic MFCs is proposed by the authors of this article. We found it interesting how structural composition, material choice and maintenance operation for the targeted MFCs can differ depending on accessibility to modify the anode burying sediments.
- Line 344-346 - remove anodes, modified cathodes since the authors are describing the materials and not the type of electrodes where they can be encounteres.
We have rephrased the statement as following:
“There are reports of using multiple engineered configuration of carbon materials like activated carbon fiber felt [140], granulated activated carbon (GAC) [114,141], modified Polyaniline- graphene nanosheets (PANI‐GNS) [126] and composite multi wall carbon nanotube [61,139,142] materials.”
- Line 347 – what the authors mean by small, medium and large? Is this size? And what is the size range of each type of electrodes? This should be state in the text.
We have added the range and the updated statement is following:
“Benthic MFC electrodes can be of small [11,137] (within 4 meter) , medium [125,143], (between 5 to 8 meters) and large (above 8 meters) [136].”
- Line 352-355 – the sentence is not clear. Rephrase it.
We have added two additional references to support the statement:
“Distributed benthic MFCs are considered to be a practical solution for harsh marine environment. This is due to the fact in case the failure of one MFC anode/ cathode, the other electrodes still remain operational and thus provide enhanced durability [65,115,118,129,141,144].”
- Line 366 - Section 3.2.2.2. describe Floating MFCs, but this section only describe floating cathodes. This should be explained better, to not confuse the reader.
We agree with the reviewer our previous version could be misleading for readers. Therefore we added an explanatory statement in the first paragraph of the sub-section to indicate that floating MFCs are in fact floating cathode MFCs:
“Thus for high depth configurations, sediment MFCs whose cathodes are submerged under water might not be optimal [151] and for such cases floating cathode MFCs or in short floating MFCs emerged as a viable solution.”
- Line 378 – What RR means?
From literature, we found that in the textile industry Scarlet RR is a disperse dye extensively used for dyeing polyester fibers but did not find the meaning of the abbreviation. We have included the explanation in the text accordingly:
“a disperse dye extensively used for dyeing polyester fibers in textile industry [147]”
- Line 425 – what oxidization mean?
By oxidization, in general, we referred to catabolic processes of the organic compounds (e.g., respiration), by aerobic organisms, and for clarification we have rephrased the section as below:
“These electrochemically active bacteria are capable of exocellular electron transfer either by having conductive pili or by generating electroactive proteins or molecules that work as natural mediators [53,55] during their catabolic activities for obtaining energy stored in the biomass.”
- Line 477 – oxygen is with a lowercase letter
We have updated the sentence accordingly.
- Line 527 – remove “opt to”
We have updated the sentence accordingly.
- Line 530-531 and Line 554-557 – these two sentences say the something. Rephrase them or remove one.
We agree that there has been a repetition and we have removed the first two lines.
- Line 533-534 – the sentence does not read well. Rephrase it.
We have rephrased the description accordingly:
“These sensors mostly measure temperature[11,23,28], levels of the phreatic aquifers [161] or pH level [201]. These applications require low power and still providing power output of this order with MFC systems is challenging. This is related to the fact that voltage reversal occur while stacking MFC cells and every MFC based system require customized power management units.”
- Line 539 – substitute “he” by “they”
We have updated the sentence accordingly.
- Line 542 – the system is repeated
We have re-phrased the sytem into :
“generate micro-watt range power in both continuous and burst mode.”
- Line 561 – remove “use of”
We have updated the sentence accordingly.
- Line 619 – The numbers in the elements should be superscript.
We have updated the sentence accordingly.
- Line 653-654 – This sentence is not clear: EBFC or GBFC are the second largest group…? But both are, or just one? The authors should explain better their differences, and why there is a need to have two different names.
EBFC and GBFC both refer to the same class. In order to clarify the ambiguous meaning, we have mentioned that EBFCs are also referred to as GBFCs and from then on only used EBFC as the class name.
- Line 665-667 should appear first in this section, and then describe the differences between the glucose BFC and the others.
We have re-ordered the paragraph in page 17 according to the reviewer’s suggestions.
- Line 668 – are the authors describing the Glucose BFC or just Enzyme BFC? This is clear and should be modified.
Our objective is to describe enzyme based bio fuel cells which are (also a class/ type of) glucose based biofuel cells. For clarification we have added the following statement in page 20, line 828:
“and are also referred to as glucose biofuel cells”
- Line 711 – What in vitro EBFC means?? Is there an in vivo EBFC?
By the term ‘in vitro EBFCs’ we wanted to indicate those where the EBFC function was tested outside the living organisms and there are also ‘in vitro EBFCs’ which have been implanted inside or outside living animals. For
better clarification we have included the following phrase in page 910:
“i.e., EBFCs which have been tested outside any living organism and in a lab environment”
- Line 713 – What PQQ in PQQ-dependent glucose dehydrogenase means?
We have added the following explanatory phrase with reference:
“which is a quinoprotein enzyme with pyrroloquinoline quinone (PQQ) cofactor for facilitating glucose oxidation [259],”

Reviewer 3 Report
The authors were able to give an exhaustive outlook on bioelectrochemical systems, that is not an easy task given the continuous signs of progress and evolution in the field. The review is of particular interest for the readers because it considers all BESs, including the Enzyme Based Fuel Cells and biomechanical energy harvesting technologies. The paper is well written and organized, with a satisfactory number of references. I, therefore, recommend publishing it after solving a few minor issues, listed below.
- The name of organisms should be in Italic (see line 447, for example)
- Line 165: what does the words "and does not yet exist" refer to?
- Lines 177-178: Please rephrase: "However, examples of additional mechanical aeration procedures are also widely reported which require additional cost components"
- Lines 180-184: the period is too long: please consider revising the grammar form/punctuation.
- Line 206: "inhibit of oxygen diffusion": it should be "inhibit oxygen diffusion" or "inhibition of oxygen diffusion"
- Line 257: it's " a continuous output" and not "an continuous output"
- Line 326: it should be linked to line 325
- Lines 382-384: the period is too long and with a weak grammar structure
- In paragraph "3.2.2.3. Terrestrial MFCs ", line 391, the authors talk about the MFCs applied to soil treatment. The authors report many examples of applications but there are a few missing: the potential integration of MFCs to green infrastructures, plant-MFCs to produce electric power on rooftops and, again, plant MFCs for future smart agriculture. See the following references:
-
- Theodore Endreny, Claudio Avignone-Rossa, Rosa Anna Nastro. (2020). Generating electricity with urban green infrastructure microbial fuel cells. Journal of Cleaner Production, 263, 2020, 121337;
- Chung-Yu Guan, Chang-Ping Yu, Evaluation of plant microbial fuel cells for urban green roofs in a subtropical metropolis, Science of The Total Environment, 2020, 142786;
- Davide Brunelli, Pietro Tosato, Maurizio Rossi, Flora Health Wireless Monitoring with Plant-Microbial Fuel Cell, Procedia Engineering, 168,2016, pp 1646-1650.
The authors talk extensively about SMFCs as a source of energy from sediments, but they do not almost mention about the utilization of the same devices to the remediation of sediments polluted by hydrocarbons and metals, even though this is one of the most investigated utilization of SMFCs. I suggest adding a section or at least a few lines about this topic. See, for example:
- Wen-Wei L, Han-Qing Y. (2015) Stimulating sediment bioremediation with benthic microbial fuel cells. Biotechnol Adv 33 (2015) 1–12.
- Nastro RA, Gambino E., Toscanesi M., Arienzo M., Ferrara L., Trifuoggi M.. 2019. Microbial Fuel Cells Remediation Activity of Marine Sediments Sampled at a Dismissed Industrial Site: What Opportunities? Journal of Cleaner Production. 235: 1559-1566.. Doi: https://doi.org/10.1016/j.jclepro.2019.07.0190959-6526.
- Li H, Tian Y, Qu Y, Qiu Y, Liu J, Feng Y (2017) A Pilot-scale Benthic Microbial Electrochemical System (BMES) for Enhanced Organic Removal in Sediment Restoration. Sci ReP 7:39802 DOI: 10.1038/srep39802.
Period in line 709 seems incomplete or should be followed by a list of EBFCs
There are some double spaces among words within the text.
Author Response
Reviewer #3:
The authors were able to give an exhaustive outlook on bioelectrochemical systems, that is not an easy task given the continuous signs of progress and evolution in the field. The review is of particular interest for the readers because it considers all BESs, including the Enzyme Based Fuel Cells and biomechanical energy harvesting technologies. The paper is well written and organized, with a satisfactory number of references. I, therefore, recommend publishing it after solving a few minor issues, listed below.:
We would like to thank reviewer for his/her availability to review our manuscript and provide detailed instructions on how to improve the manuscript.
Outlined comments
- The name of organisms should be in Italic (see line447, for example)
We have corrected the name of organisms in italic for the suggested ones :
- Line 489: “Saccharomyces cerevisiae”
- Line 494: “Candida melibiosica”
- Line 509: “Kluyveromyces marxianus”
- Line 516: “Scedosporium dehoogii”
- Line 524: “Trametes versicolor”
- Line 524: “Shewanella oneidensis”
- Line 165: what does the words "and does not yet exist" refer to?
For clarification we have rephrased the statement in line 176-177 as follows –
“although such an optimal MFC cathode configuration has yet not been achieved.”
- Lines 177-178: Please rephrase: "However, examples of additional mechanical aeration procedures are also widely reported which require additional cost components"
We have rephrased the indicated statement in line 196 as follows –
“While multiple examples of mechanical aeration procedures are reported, use of such units require additional cost [64,70–72]”
- Lines 180-184: the period is too long: please consider revising the grammar form/punctuation.
We agree with your suggestion and have rephrased the sentence in line 199 - 205 accordingly –
“Air cathodes [70] have emerged as popular low cost solution providing sustainable aeration at cathodes. In this configuration the cathode surface partially remains open to air and continuously receives oxygen supply. Janicek et al. [76] reported a generalized air cathode configuration which consists of a catalyst layer that faces solution side of the cathode, a gas diffusion layer that faces air, and a conductive base material layer. The conductive layer also acts as a current collector as well as a mechanical support provider.”
- Line 206: "inhibit of oxygen diffusion": it should be" inhibit oxygen diffusion" or "inhibition of oxygen diffusion"
We have updated the phrase in line 223 accordingly –
“Inhibit oxygen diffusion”
- Line 257: it's " a continuous output" and not "an continuous output"
We have updated the phrase in line 628 accordingly –
“a continuous output”
- Line 326: it should be linked to line 325
We have rephrased and merged the statement in line 341 according to your suggestions:
- “Degradation of electrode materials quality due to –
- electrochemical deposition,
- corrosion,
- impacts of water flow,
- fish grazing [128] and
- Burrowing of anodes [65]. ”
8. Lines 382-384: the period is too long and with a weak grammar structure.
We agree with your suggestion and have rephrased the sentence in line 407-410 accordingly –
“A series of floating MFC configurations were reported by Schievano et al. [150,151]. Applicability for one the two [150] was tested both in wastewater as well as in a natural water body. The second one’s [151] performance was tested for powering remote sensors and wireless data transmission.”
- In paragraph "3.2.2.3. Terrestrial MFCs ", line 391,the authors talk about the MFCs applied to soil treatment. The authors report many examples of applications but there are a few missing: the potential integration of MFCs to green infrastructures, plant-MFCs to produce electric power on rooftops and, again, plant MFCs for future smart agriculture. See the following references.
We thank the reviewer for guiding us to include an important new applicability of the terrestrial MFCs and we have included the following text for incorporating the suggestion in line 430-436 –
“Another promising and emerging concept is integration of MFCs with green infrastructures for achieving cleaner environment in urban landscape[157,158]. The green roofs can be of particular interest [158–160] for this purpose since in addition to electricity generation, they also provide cooling impact and thus reduce additional energy demand. An additional smart agricultural application should also be noted where MFC integrated power generation systems are utilized for plant heath monitoring applications[161,162]”
- The authors talk extensively about SMFCs as a source of energy from sediments, but they do not almost mention about the utilization of the same devices to the remediation of sediments polluted by hydrocarbons and metals, even though this is one of the most investigated utilization of SMFCs. I suggest adding a section or at least a few lines about this topic. See, for example:
This is again a very useful guiding instruction for us and we thank the reviewer. We have added a new application section in the article as follows which comprises of the suggested literature and some more:
“3.4.3 Bioremediation
An additional application of MFC technology has been bioremediation and also one of the most investigated applications of sediment MFCs [233]. In this process microorganisms are utilized to treat polluted to sites to break down environmental pollutants, to regain back their original condition [234]. This has long remained as an alternative natural process of waste removal from land [235]. For more details on recent developments on bioremediation of sediments, please refer to the works of [236]. There have been particular in removing in removing organic [237] hydrocarbon [238] and metal [239] based pollutants .
In their review, Dominguez-Benetton et al. [240] summarized the latest mechanisms of metal recovery using microbial fuel cells. In another review, Wang et al. [229] classified mechanisms of metal recovery using MFCs in 4 different categories: direct metal recovery with abiotic cathodes; metal recovery with externally powered abiotic cathodes, metal conversion with bio-cathodes and metal conversion with externally powered bio-cathodes supplemented by external power sources.”
- Period in line 709 seems incomplete or should be followed by a list of EBFCs
Indeed we have did not complete the classification at the mentioned line by mistake. We have added the following section to complete the sentence.
“We can classify EBFCs into four major groups: In-vitro, plant powered, animal powered and human powered. Animal power EBFCs can be further classified into externally and internally implantable sub-groups. Nearly all presently available human powered EBFCs fall under the wearable sub-group. This classification is depicted in Figure 12.”
- There are some double spaces among words within the text.
Thank you for identifying the grammatical error. We have corrected the double spaces in multiple lines.

Round 2
Reviewer 2 Report
See attached file.

Author Response
Reviewer #2:
In the revised manuscript “Towards Bio-Hybrid Energy Harvesting in the Real-World: Technologies and Strategies Pushing the Boundaries of Bio-Electrochemical and Bio-mechanic processes” the authors describe the recent progresses on three different bio-energy harvesting systems including microbial fuel cells, enzyme based fuel cells and biomechanical energy harvesters. I consider that the revised version of the manuscript is improved relative to the previous version. Although the authors took into account the comments and suggestion of the reviewers, improving the manuscript, there are still small changes that need to be performed before the manuscript is accepted for publication:
We would like to thank reviewer for his/her availability to review our manuscript and guiding us thoroughly for improvement of the article. Below are the reviewer’s comments and their corresponding replies.
Outlined comments
- There are still abbreviations that are still necessary to be modified in the text, in particular MFC that appear as MFC and microbial fuel cells. Modify to MFC: line 71,79, 163, 230, 305, 420, 455, 519, 523, 551, 579, 613, 670, 763, 843, 1081, 1083
- And PMS: line 644, 649, 651
- Remove MPPT – line 645 and 1146
- Remove constructed wetland MFC – Line 569
- Remove PHB – Line 580
- Remove CDC – Line 730
- Add TENG in line 1015 and replace it in 1021. Remove the full name from line 1026, 1052, 1057
- Remove MPPT - line 1146
Updated accordingly.
- Line 93 – replace nutrition by energy
Incorporated the suggestion according to the following:
“In a classic configuration, the microorganisms decompose the organic substrate in the anaerobic anode chamber through catabolic processes to obtain energy nutrition and generate electrons and protons/ cations as by product.”
- Line 112 – remove e, from thee.
Updated accordingly.
- Line 126 - … first organisms shown to …
Incorporated the suggestion according to the following:
“Geobacter sulfurreducens was one of the first organisms shown to produce conductive nanowires.”
- Line 128 – pilA
Updated accordingly.
- Line 128 – insert reference in this sentence.
Incorporated the suggestion according to the following:
“These nanowires were latter referred to conductive pilA [53]”
- Line 129 – … their outer-membrane, allowing the transfer of electrons …
Incorporated the suggestion according to the following:
“as extensions of their outer-membrane, allowing the transfer of electrons to anode without requiring soluble electron shuttles.”
- Rearrange sentences 131-136. Put as first sentence line 133-136, and then line 131-133.
Incorporated the suggestion according to the following:
“The presence of bacterial biofilm can be highly advantageous for MFCs because of their electroactive nature aids to generate electricity more efficiently. This biofilm is a complex, organic, polymeric matrix, produced by the bacteria themselves at any biotic or abiotic surface and these organic films can be formed by a single bacterial (pure-culture) or multiple bacterial species (mixed-culture) [53].”
- Line 151 – it is not only wastewater environment, it can also be other wastes, electrolytes. Re-write this sentence.
“…withstand wastewater environment with diverse organic and inorganic contents [61], other wastes, electrolytes and soil contaminating components.”
- Line 316 – …increased expense. Biofouling …
Incorporated the suggestion according to the following:
“…and increased expense. Bio-fouling and clogging…”
- Line 334 – … therefore we refer them …
Incorporated the suggestion according to the following:
“…buried anodes and therefore them we refer them in our classification…”
- Line 346 – Burrowing anodes
Updated accordingly at line 303.
- Line 408-409 – the sentences are not clear. For example, modify ‘for one the two’ and the ‘second one’s’ to ‘Applicability of one system…’, and ‘In another system, performance …’, respectively.
Incorporated the suggestion according to the following:
“Applicability for one the two of the first configuration [156] was tested both in wastewater as well as in a natural water body. The second one’s Performance of the second configuration [157] was tested…”
- Line 467 – MFCs
Updated accordingly.
- Line 472 – remove is
Updated accordingly.
- Line 479 and 480 – the species names are in italic
Incorporated the suggestion according to the following:
“Two most studied exoelectrogen bacteria for MFCs are Geobacter sulfurreducens [54,174] and Shewanella oneidensis MR-1 [54,175,176].”
- Line 481 – the extracellular pilin can be by pili formed by PilA or appendages formed by the cytochrome OmcS (Wang 2019 - doi.org/10.1016/j.cell.2019.03.029). Modify the sentence
Incorporated the suggestion according to the following:
“either through the extracellular pilin can be by pili formed by PilA or appendages formed by the cytochrome OmcS [177].”
- Line 485 – c is in italic
Updated accordingly.
- Line 485 – nanowires in Shewanella are extensions of the membrane and therefore the direct electron transfer is also performed by OM cytochromes.
Incorporated the suggestion according to the following at line 442:
“It should be noted that nanowires in Shewanella are extensions of the membrane and therefore the direct electron transfer is also performed by OM cytochromes.”
- Line 489 – the sentence is not clear
Incorporated the suggestion according to the following at line 448:
“…where Saccharomyces cerevisiae was tested as MFC biocatalyst and their performance in the MFC setup was tested without any mediator and in three different pH conditions.”
- Line 490 – the sentence is not clear
Incorporated the suggestion according to the following at line 450:
“The test results indicated that Saccharomyces cerevisiae can be used as an effective anodic biocatalyst component for MFC setups.”
- Line 507 – the sentence is not clear
Incorporated the suggestion according to the following at line 466:
“Kaneshiro et al. [46] reported examining MFC power generation performance using for seven different kinds of yeasts. Their performances were evaluated with a milliliter scale, dual-chamber MFC cell configuration with where carbon fiber bundles were used as electrodes.”
- Line 511 – what is woody biomass ?
Incorporated the suggestion according to the following at line 469:
“For substrate they utilized a combination of glucose and xylose, which was extracted from wood biomass.”
- Line 513-514 – the sentence is not clear
Updated the sentences according to the following at line 474-475:
“Impact of two inorganic mediators, employing two different mediators, methylene blue and methyl red were tested on a Fungi-MFC and was reported by Christwardana et al. [49].”
- Line 514 – insert final dot after the reference
Updated accordingly.
- Line 516 – replace ‘based’ by ‘in a’
Updated the sentences according to the following at line 477-478:
“a fungus Scedosporium dehoogii based in a MFC”
- Line 519 – power output
Updated the sentences according to the following at line 480:
“provided a stable power output of 50 mW/m2 .”
- Line 529 – The yeast species name should be in italic
Updated the sentences according to the following at line 490:
“…Klebsiella pneumonia boosted up the performance of yeast Lipomyces starkeyi based MFCs.”
- Line 552 – species name in italic
Updated the sentences according to the following at line 513:
“for bio-electricity generation using Chlorella vulgaris in the cathode.”
- Line 556 – remove (2015)
Updated accordingly at line 516.
- Line 562 – remove are
Updated accordingly at line 522.
- Line 566 . remove comma
Updated accordingly at line 526.
- Line 572-573 – the authors are describing bacteria, when in this section they should describe Algae or plant aided as described in Figure 8.
We wanted to stress of the fact that photo-reactor aided MFCs influenced developments of bio-cathodes. Moreover, the fact that bacterial growth on cathodes can also boost MFC performance is considered as an additional advantage for biocathodes. For this reason, we have added another phrase before the indicated sentence.
“In addition to the finding that phototrophic organisms grown on cathodes can provide additional oxygen supply [190], the finding that bacteria can also be used as biocatalysts to accept electrons from the cathode electrode emphasized development of bio-cathodes [203].”
- Rearrange section 3.4.1. Put sentence 620-623 in the beginning of the section, and remove first sentence – line 607, to make this section clearer.
Incorporated the suggestions accordingly:
“Lab-scale MFCs, which are in the order of milliliters in volume [103] generate electricity in the range of mW/m2 power density [11,104,105] and around mA/m2 in current density [106]. Up until now, most of the dedicated PMSs power management systems have been developed for marine remote sensors with low power requirements [107].
One of the major applications of MFC generated electricity is powering up remote, low voltage marine sensors. MFCs are widely used for powering marine These sensors which mostly measure temperature[11,23,28], levels of the phreatic aquifers [167] or pH level [210].”
- Line 632 – its
Updated accordingly at line 587.
- Line 679 – of
Updated accordingly at line 634.
- Line 692 – whether
Updated accordingly at line 646.
- Line 699 – reported the use of …
Updated accordingly at line 653.
- Line 703 - -1 is in superscript
Updated accordingly at line 657.
- Line 706-707 – the mention of (a) and (b) is not clear. Remove it
Updated the sentences according to the following at line 659-661:
“This study offered two major observations: (a) and (b) the photosynthetic activity of the macrophytes, Phragmites australis was dependent affected on the light/darkness changes and this caused voltage fluctuations and affected stable performance of the system.”
- Line 718 – replace beta by -
- Line 735 – combination of MFC
Updated accordingly at line 688
- Line 736 and 744 – remove cells in microbial electrosynthesis
Updated accordingly at line 689, 697.
- Line 741 – insert correct reference
Updated the sentences according to the following at line 707-708:
“MFCs have been reported to recover Ag(I), Au(III), Co(II), Cd(II), Cr(VI), 707 Cu(II), Hg(II), Pb(II), Se(IV), V(V), U(VI), and Zn(II) [30][Nancharaiah et al (2015)].”
- Line 742 – wastewater
Updated accordingly at line 710.
- Line 743-745 should be move to section 3.4.3.
Reorganized.
- Line 772 – …cathode. They have used …
Updated accordingly at line 729
“…double air cathode. They have used cow…”
- Line 847 – implementation
Updated accordingly at line 790
- Line 849-851 – can be removed since it is a repetition
Removed.
- Line 851-854 should be combined with sentence 836-837
Updated accordingly at line
“EBFCs are bio-electrochemical cells that can extract energy from glucose and alcohol based organic substances [259]. In a classic EBFC configuration glucose oxidase or glucose dehydrogenase (GDH) are immobilized at the bio-anode for glucose oxidation while oxygen is reduced at biocathode using immobilized laccase or bilirubin oxidase and generate power [33].”
- Line 866 – remove ‘till now’
Updated accordingly at line 807.
- Line 875 – hypothesize that these
Updated accordingly at line 813-814.
- Line 948 – there is no need for this subsection. I suggest to remove it.
We removed the subs-section. We wanted to give readers an introductory idea about host immune response, so we utilized part of the text and the figure within section 4.1.
- Line 954 – remove causeing
Removed.
- Line 962 – 964 – should be place previously in the text, and not in the limitation of these devices
Removed and replaced at line as follows:
“A new group of EBFCs are now being investigated which can be implanted externally. For example, Toit et al. reported power generation from transdermal extract of pig skin [287].
Scopes of utilization for such external or wearable EBFCs for human use are also being explored. For example, studies on use of enzymatic biofuel cell on a contact lens [281–283] and patches [284,285] have been reported. Generation of electrical power from human perspiration are also reported [13,286].”
- Line 992 – Piezoelectric
Updated accordingly
- Line 1059 – replace ; by and
Updated accordingly at line 999.
- Line 1066 – the species name should be in italic - Cotinis nitida (Linnaeus)
Updated accordingly at line 1012.
- Line 1224 - energy
Updated accordingly at line 1130, 1132.
- The name of the organisms in the references should be in italic.
Updated.
